# RNF213 regulates blood–brain barrier integrity by targeting TRAF3 for type I interferon activation during *A. baumannii* infection

**Yanfeng Li[1‡], Qingqing Xie[1‡], Luyu Yang[1], Qian Jiang[2,3], Zhiping Liu[2], Chengjiang Gao[4,5], Xiaopeng Qi[1,6*], Tao Xu[1*]**

**1** Key Laboratory for Experimental Teratology of the Ministry of Education, Advanced Medical Research Institute, Qilu Hospital, Cheeloo College of Medicine, Shandong University, Jinan, China, **2** School of Nursing; Gannan Medical University; Ganzhou, Jiangxi, China, **3** School of Graduate, China Medical University, Shenyang, Liaoning, China, **4** Department of Immunology and Key Laboratory of Infection and Immunity of Shandong Province & Key Laboratory for Experimental Teratology of Ministry of Education, Shandong University, Jinan, China, **5** Department of Immunology, School of Basic Medical Sciences, Shandong University, Jinan, China, **6** State Key Laboratory for Innovation and Transformation of Luobing Theory; The Key Laboratory of Cardiovascular Remodeling and Function Research, Chinese Ministry of Education, Chinese National Health Commission and Chinese Academy of Medical Sciences, China

‡ These authors share first authorship on this work.
* xqi@email.sdu.edu.cn (XQ); tao.xu@sdu.edu.cn (TX)

## Abstract

RNF213 is the first identified susceptibility gene for moyamoya disease, and the encoded protein was recently recognized as a key antimicrobial protein. However, the function of RNF213 in host defense against brain infection remains unclear. Here, we show that increased expression of *Rnf213* is significantly regulated by interferon alpha/beta receptor (IFNAR) signaling during bacterial infection and ligand stimulation. RNF213 deficiency impairs type I interferon (IFN-I) production and decreases the expression of interferon-stimulated genes (ISGs) in response to IFN-β stimulation and *Acinetobacter baumannii* infection. Mechanistically, RNF213 interacts with TRAF3 and mediates the K27-linked polyubiquitination of TRAF3 at K160. RNF213 regulates the expression of the endothelial tight junction-related genes *Claudin-5, Occludin, and Pecam1* via IFN-I signaling. Furthermore, RNF213 deficiency in non-immune cells increases blood–brain barrier (BBB) disruption and the bacterial load in the brain parenchyma in response to *A. baumannii* infection due to impaired IFN-I signaling. Thus, RNF213 mediates BBB integrity by targeting TRAF3 for the regulation of IFN-I signaling against bacterial brain infection. Our study principally provides a deeper understanding of the function of RNF213 and reveals potential therapeutic targets against bacterial brain infection and moyamoya disease.

**Data availability statement:** All relevant data are with in the manuscript and its Supporting Information files.

**Funding:** This work was supported by the National Natural Science Foundation of China (82472293 to TX, 82125021 to XQ, and 82321002 to CG), Cutting Edge Development Fund of Advanced Medical Research Institute (GYY2023QY01 to XQ), and Shandong Province (2022GJJLJRC02-005 to XQ). The funders had no role in study design, data collection and analysis, decision to publish, or preparation of the manuscript.

**Competing interests:** The authors have declared that no competing interests exist.

## Author summary

The destruction of the blood-brain barrier (BBB) by different bacteria or systemic inflammation can lead to aggravate various central nervous system (CNS) diseases. Studies have shown that the BBB damage in Moyamoya disease (MMD) patients is significantly greater than in patients with other cerebrovascular diseases. The function of RNF213, the first identified susceptibility gene for MMD, remains unclear in host defense against brain infection. The present study found that RNF213 regulates type I interferon (IFN-I) production in response to *Acinetobacter baumannii* infection by mediating the ubiquitination of TRAF3. Moreover, RNF213 regulates the expression of the endothelial tight junction-related genes via IFN-I signaling. We provide evidence that RNF213 mediates BBB integrity by targeting TRAF3 to regulate IFN-I signaling against bacterial brain infection. Our data reveal a previously unrecognized target by which RNF213 regulates the IFN-I response and establishes a link between this MMD susceptibility gene and infectious diseases.

## Introduction

As the first line of host defense against pathogen infection and autoimmune diseases, innate immunity is rapidly activated upon the detection of invading pathogens and plays an important role in host defense [1,2]. Type I interferons (IFN-Is), which bind to the IFN-α/β receptor (IFNAR) in both autocrine and paracrine manners to initiate a series of signaling events, are multifunctional cytokines that function as key components of the innate immune response to infection [3–5]. Thus, the secretion of IFN-I and the consequent downstream events are tightly regulated. Toll-like receptors (TLRs) are important molecules of the innate immune system, as they detect conserved structures in a wide range of pathogens, triggering innate immune responses. The activation of membrane-bound TLR4 and endosomal TLR3 can induce TIR domain-containing adaptor-inducing interferon-β (TRIF)-dependent IFN-I production [6,7]. *Acinetobacter baumannii* (*A. baumannii*) is a multidrug-resistant (MDR), common opportunistic, gram-negative coccobacillus that exhibits extensive resistance to antibiotics and enters through skin and airway defects in hospitalized and vulnerable patients [8,9]. TRIF-dependent IFN-I signaling and the related downstream molecules are essential for *A. baumannii* infection-induced pyroptosis and necroptosis, which play positive roles in host defense against *A. baumannii* pulmonary infection [10]. The incidence of *Acinetobacter baumannii* meningitis has increased recently due to the emergence of MDR and extensively drug-resistant (XDR) *A. baumannii* strains [11,12]. However, the molecular mechanisms underlying the host defense against *A. baumannii* brain infection remain largely unclear.

Mouse RING finger protein 213 (RNF213), which possesses an AAA$^+$ ATPase domain and an E3 ubiquitin ligase domain, is a giant E3 ubiquitin protein ligase consisting of 5,207 amino acids and is associated with autoimmunity, autophagy,

angiogenesis and lipid metabolism [13–15]. Recently, several studies on RNF213 have revealed its anti-infective function. RNF213 inhibits the cytoplasmic proliferation of *Salmonella* and is essential for the production of the bacterial ubiquitin coat through both direct and indirect mechanisms [16]. In addition, RNF213 has been demonstrated to restrict the de novo infection and lytic reactivation of Kaposi's sarcoma-associated herpesvirus (KSHV) by degrading Replication and Transcription Activator (RTA) through the ubiquitin–proteasome pathway [17]. RNF213 has also been identified as an ISG15-binding protein and a cellular sensor of ISGylated proteins to counteract broad-spectrum antimicrobial activity against *Listeria monocytogenes*, HSV1, RSV and CVB3 [18].

Moyamoya disease (MMD) is an uncommon chronic cerebrovascular disease characterized by progressive stenosis or occlusion at the terminal portion of the internal carotid artery and the initial segment of the middle cerebral artery or the anterior cerebral artery, along with abnormal vascular network development at the base of the brain [19–21]. The blood–brain barrier (BBB) is formed mainly by specialized brain microvascular endothelial cells (BMECs) and is sealed by continuous complexes of tight junctions and ensheathed by astrocytes and pericytes [22]. The BBB is a highly selective barrier that protects the central nervous system (CNS) from the unwanted entry of pathogens, macromolecules and cells. Studies have shown that the degree of BBB damage in MMD patients is significantly greater than that in patients with other cerebrovascular diseases [23]. RNF213 is recognized as an important and common susceptibility gene for MMD [24]. Genetic variants of RNF213 are associated with a variety of autoimmune diseases and with MMD. Multiple missenses mutations of RNF213 associated with MMD patients cluster significantly in the C-terminal region encompassing RING domain, which greatly reduce the ubiquitin ligase activity of RNF213 [25–27]. The identification of RNF213 as an immune sensor revealed a causal link between MMD and infection [28]. However, the mechanism by which RNF213 mutations affect the pathogenesis of MMD remains unknown.

In this study, we demonstrate that the expression of RNF213 is significantly induced by *A. baumannii* infection and that RNF213 positively regulates host defense. *Rnf213*-deficient mouse skin fibroblasts (MSFs) and BMECs exhibit impaired IFN-I signaling. RNF213 modulates the expression of endothelial tight junction (TJ)-related proteins and BBB integrity via IFN-I signaling. Furthermore, RNF213 increases TRAF3-mediated IFN-I signaling by mediating the K27-linked ubiquitination of TRAF3 at K160. Taken together, our findings identify RNF213 as a positive regulator of the IFN-I signaling pathway and indicate that RNF213 plays important roles in maintaining BBB integrity during *A. baumannii* infection.

## Results

### RNF213 expression is induced in response to pathogenic infection and stimuli

To explore the critical regulators involved in the IFN-I signaling-mediated ubiquitination pathway during bacterial infection, we performed genome-wide RNA sequencing (RNA-seq) analysis of WT and IFNAR-deficient (*Ifnar*$^{-/-}$) bone marrow-derived macrophages (BMDMs) following infection with *A. baumannii.* The expression of 127 and 401 genes was significantly greater (≥2-fold) in WT BMDMs than in *Ifnar*$^{-/-}$ BMDMs under normal and *A. baumannii*-infected conditions, respectively (S1A and S1B Fig and S1 Table). Among the genes whose expression was upregulated in WT BMDMs compared with *Ifnar*$^{-/-}$ BMDMs, 93 genes were common to the normal and *A. baumannii*-infected groups (S1B Fig). Gene Ontology (GO) analysis revealed that the top 5 signaling pathways enriched in these 93 upregulated genes were associated primarily with cellular antiviral immune responses and the regulation of the innate immune response (S1B Fig). Among the 93 commonly expressed genes, *Rnf213* exhibited comparatively high expression at both basal and *A. baumannii*-infected conditions, and the expression of *Rnf213* was significantly regulated by IFNAR signaling (Figs 1A and S1C).

To determine whether the IFN-I-mediated expression of *Rnf213* was specific to *A. baumannii* infection, we examined *Rnf213* expression in WT and *Ifnar*$^{-/-}$ BMDMs and MSFs in response to infections with multiple pathogens and multiple stimuli via real-time quantitative polymerase chain reaction (RT–qPCR) analysis. The expression of *Rnf213* was markedly increased in WT BMDMs in response to infection with *A. baumannii*, herpes simplex virus type 1 (HSV-1) and rabies virus (RABV) and in response to treatment with LPS, poly(I:C) and IFN-β (Fig 1B). The induction of *Rnf213* expression was

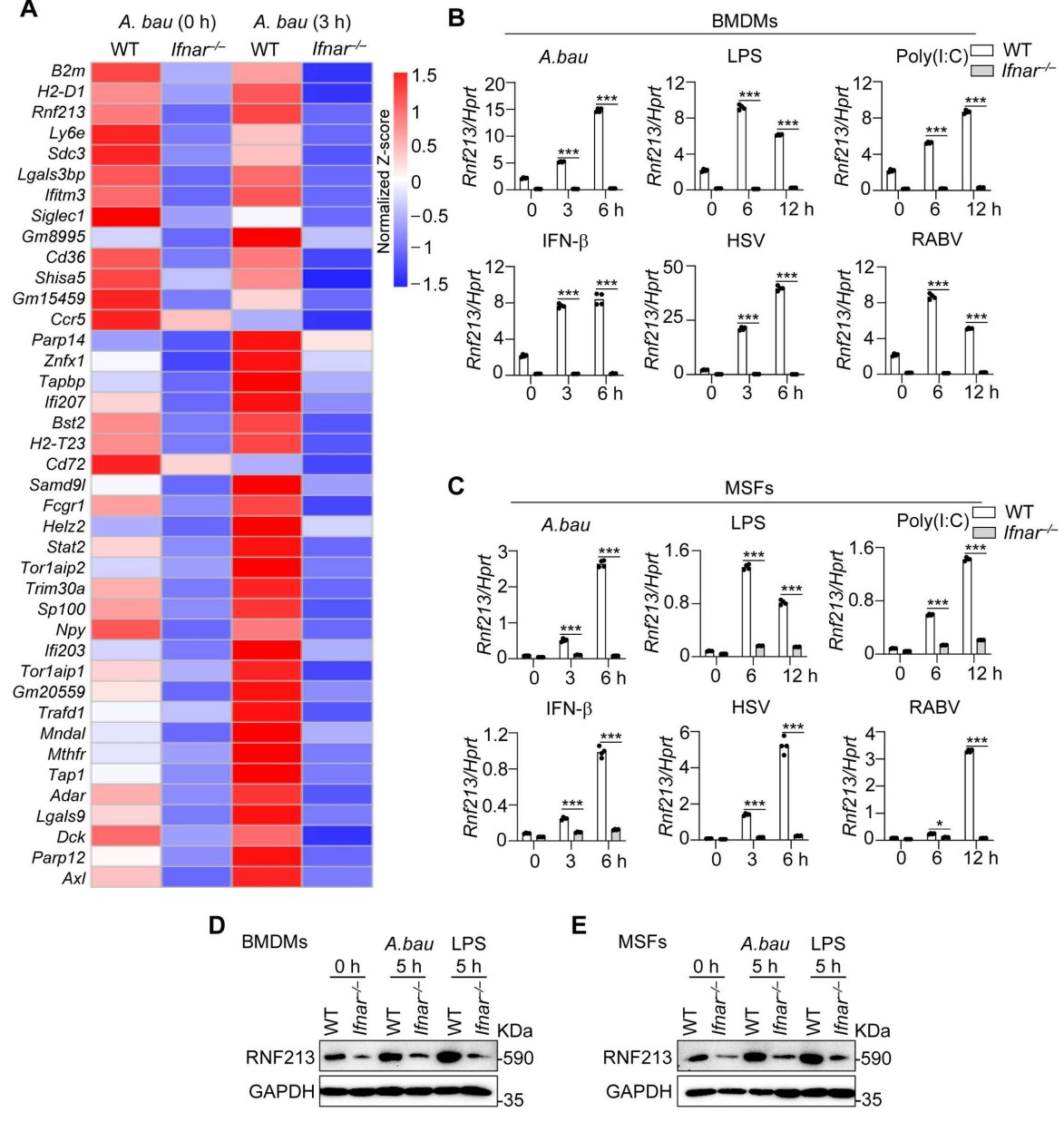

**Fig 1. IFNAR mediates *Rnf213* expression induced by pathogenic infection and stimuli. (A)** RNA-seq analysis of the gene expression of WT and *Ifnar*−/− BMDMs infected with *A. baumannii* for 3 hours. The heatmap illustrates expression patterns of the top 40 ranked genes of WT versus *Ifnar*−/− BMDMs in both uninfected and *A. baumannii*-infected conditions. Each row represents a gene, and each column represents a sample. The expression values of each gene were standardized (z-score normalization) across samples to facilitate comparison. The color blue indicates lower expression, white represents median expression, and red indicates higher expression. **(B-C)** qRT-PCR analysis of *Rnf213* in WT and *Ifnar*−/− BMDMs (B) and MSFs (C) without treatment or infected with *A. baumannii* (50 MOI), HSV-1 (3 MOI), and RABV (5 MOI); or stimulated with LPS (200 ng/mL), Poly(I:C) (40 μg/mL), and IFN-β (2 ng/mL) for indicated times (n = 4 technical replicates; 3 independent experiments). **(D-E)** Immunoblot analysis of RNF213 in WT and *Ifnar*−/− BMDMs (D) and MSFs (E) without treatment or infected with *A. baumannii* (50 MOI), or stimulated with LPS (200 ng/mL) for indicated times. Housekeeping gene *Hprt* (Hypoxanthine guanine phosphoribosyltransferase) serves as an internal control. Data are representative of 3 independent experiments with similar results **(B-E)**. Data represent Mean ± SEM for **(B, C)**, 2-sided Student's *t*-test without multiple-comparisons correction, *$P < 0.05$; ***$P < 0.001$.

significantly reduced in *Ifnar*[-/-] BMDMs compared with WT BMDMs (Fig 1B). In addition, the induction of *Rnf213* expression by pathogen infection and ligand stimulation was significantly increased in mouse skin fibroblasts (MSFs) (Fig 1C). The increases in the expression of RNF213 in WT cells but not *Ifnar*[-/-] cells induced by *A. baumannii* infection and LPS stimulation were also confirmed by western blot analysis (Fig 1D and 1E). Taken together, these results indicate that IFN-I mediates the pathogen-induced upregulation of *Rnf213*, consistent with previous studies identifying *Rnf213* as an interferon-stimulated gene (ISG) [29,30].

### IFN-I signaling is impaired in *Rnf213–/–* MSFs and BMECs

To investigate whether *Rnf213* plays a role in the immune response of IFN-I signaling, we initially examined the effect of RNF213 overexpression in WT MSFs response to *A. baumannii* infection via RT–qPCR analysis. *Rnf213* overexpression significantly increased the *A. baumannii* infection-induced expression of chemokine encoding genes *Cxcl9, Cxcl10* and transcription factor *Irf1*, which are responsive genes upon IFNAR signaling activation [3,31,32]. However, *Rnf213* overexpression had no effect on the expression of inflammatory cytokines, including *Tnfa* and *Il1a* (S2A Fig). Consistently, *Rnf213* overexpression in human umbilical vein endothelial cells (HUVECs) enhanced *A. baumannii* infection-induced expression of *Cxcl9, Cxcl10* and *Irf1* (S2B Fig). To further confirm the role of RNF213 in the immune responses, we performed small interfering RNAs (siRNAs) targeting *Rnf213* to inhibit *Rnf213* expression in WT MSFs and examined inflammatory cytokine expression and IFN-I signaling. *Rnf213* knockdown significantly decreased the *A. baumannii* infection-induced expression of *Cxcl9, Cxcl10* and *Irf1* (S2C Fig). However, *Rnf213* knockdown had no effect on the mRNA expression of inflammatory cytokines, including *Tnfa* and *Il1a* (S2C Fig).

To evaluate the physiological function of RNF213, we generated *Rnf213*-deficient (*Rnf213*[-/-]) mice via CRISPR/Cas9 gene editing (S3A and S3B Fig). Brain microvascular endothelial cells (BMECs) and MSFs were isolated from both WT mice and *Rnf213*[-/-] mice and stimulated with *A. baumannii* and LPS, and gene expression was then analyzed. The increases in the expression of *Cxcl9* and *Cxcl10* induced by *A. baumannii* infection and LPS stimulation were much lower in *Rnf213*[-/-] MSFs than in WT MSFs (Fig 2A). However, the increases in the expression of *Cxcl9* and *Cxcl10* induced by IFN-β treatment were comparable between WT and *Rnf213*[-/-] MSFs (Fig 2A). Similar patterns of *A. baumannii* infection- and LPS stimulation-induced *Cxcl9* and *Cxcl10* expression were observed in BMECs (Fig 2A). In addition, ELISA revealed that the secretion of IFN-β but not that of TNF-α was much lower in *Rnf213*[-/-] MSFs and BMECs than in the corresponding WT cells under both LPS treatment and *A. baumannii* infection conditions (Fig 2B and 2C). Consistent with these findings, *Rnf213* deficiency decreased the phosphorylation of IRF3 and TBK1 in response to *A. baumannii* infection and LPS stimulation in both MSFs and BMECs (Figs 2D, 2E, S3C and S3D). However, the *A. baumannii*-induced expression of *Tnfa*, *Il1a*, and *Il6* was comparable between WT and *Rnf213*[-/-] MSFs and BMECs (S3E and S3F Fig). Taken together, our results indicate that RNF213 positively regulates IFN-I signaling possibly by mediating events upstream of IFNAR binding.

### RNF213 deficiency causes blood–brain barrier impairment and is susceptible to brain infection of *A. baumannii*

To investigate the role of RNF213 in BBB impairment and bacterial evasion in the brain during *A. baumannii* infection, WT and *Rnf213*[-/-] mice were intravenously infected with *A. baumannii*, and the bacterial loads in the brain and spleen were determined 20 hours post infection (Fig 3A). After *A. baumannii* infection, the bacterial loads in the brains of *Rnf213*[-/-] mice were markedly greater than those in the brains of WT mice (Fig 3B). However, the bacterial loads in the spleen were comparable between WT and *Rnf213*[-/-] mice (Fig 3C). We postulated that the selective increase in the bacterial load in the brain in *Rnf213*[-/-] mice was due to the compromised BBB integrity in *Rnf213*[-/-] mice.

Previous studies have indicated that RNF213 might be a key regulator of cerebral endothelial integrity and reinforced the importance of BBB integrity in the development of MMD [33]. To investigate whether RNF213 is involved in the disruption of the BBB after *A. baumannii* infection, we intravenously injected Evans blue dye (EBD) into both WT and *Rnf213*[-/-] mice and evaluated its leakage into the brain, a classic approach for evaluating BBB integrity (Fig 3A). Compared with

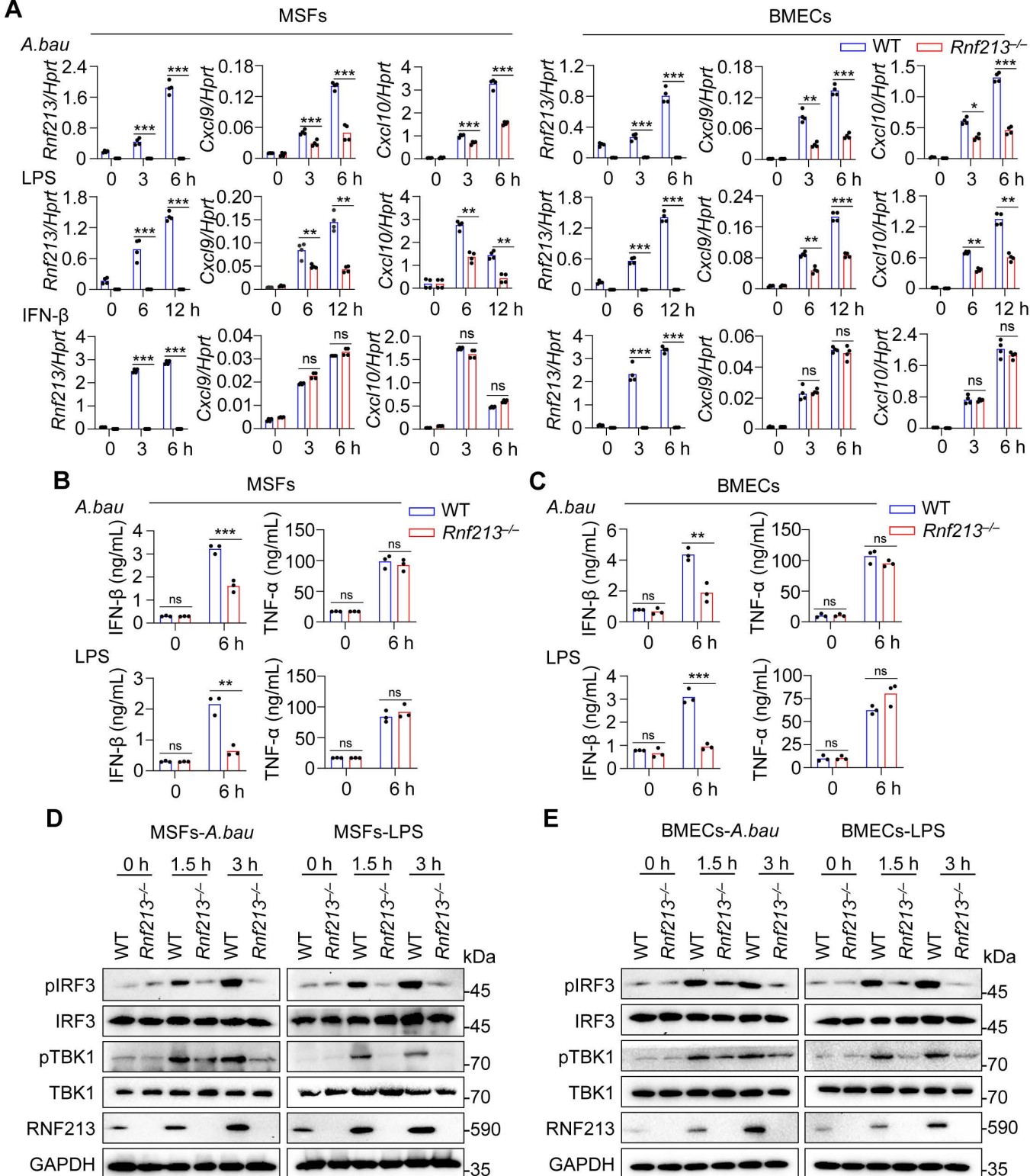

**Fig 2. RNF213 positively regulates type I IFN signaling in MSFs and BMECs.** (A) qRT-PCR analysis of *Rnf213, Cxcl9* and *Cxcl10* in WT and *Rnf213*⁻/⁻ MSFs and BMECs without treatment or infected with *A. baumannii* (50 MOI); or stimulated with LPS (200 ng/mL) and IFN-β (2 ng/mL) for

indicated times (n = 4 technical replicates; 3 independent experiments). **(B-C)** ELISA analysis of IFN-β and TNF-α release in WT and *Rnf213⁻/⁻* MSFs (B) and BMECs (C) without treatment or infected with *A. baumannii* infection (50 MOI) and stimulated with LPS (200 ng/mL) for indicated times (n = 3 biologically independent samples). **(D-E)** Immunoblot analysis of phosphorylation of IRF3, TBK1, and IRF3, TBK1 and RNF213 in WT and *Rnf213⁻/⁻* MSFs (D) and BMECs (E) without treatment or infected with *A. baumannii* (50 MOI), or stimulated with LPS (200 ng/mL) for indicated times. Data are from 3 independent experiments (B, C) or representative of 3 independent experiments with similar results (A, D, **E**). Data represent Mean ± SEM for (A, B, **C**), 2-sided Student's *t*-test without multiple-comparisons correction, *$P < 0.05$; **$P < 0.01$; ***$P < 0.001$; ns, not significant.

uninfected control mice, *A. baumannii*-infected WT mice displayed visible leakage of EBD (Fig 3D and 3E). Massive leakage of EBD caused by *A. baumannii* infection was observed in the brains of *Rnf213⁻/⁻* mice compared with WT mice (Fig 3D). Confocal microscopy further revealed that EBD fluorescence in the brain parenchyma was markedly greater in *Rnf213⁻/⁻* mice than in WT mice following *A. baumannii* infection (Fig 3E and 3F). Given that RNF213 might be a key molecule in tight junction formation, we further examined the expression of mRNAs encoding endothelial TJ-related proteins in the brain tissue of WT and *Rnf213⁻/⁻* mice. RT–qPCR analysis revealed that RNF213 deficiency inhibited the expression of *Claudin-5, Occludin*, and *Pecam1* in response to *A. baumannii* infection (Fig 3G). Consistent with these findings, immunofluorescence staining revealed decreased expression of Claudin-5 and Occludin following *A. baumannii* infection in *Rnf213⁻/⁻* mice compared with WT mice (Figs 3H, 3I, S4A and S4B).

To determine whether immune cells or nonimmune cells contribute to the BBB impairment and increased bacterial load in the brain in *Rnf213⁻/⁻* mice during *A. baumannii* infection, we performed bone marrow transplantation (BMT) between WT (CD45.1⁺) and *Rnf213⁻/⁻* (CD45.2⁺) mice (S5A Fig). To examine the engraftment efficiency, we performed flow cytometry analysis and found that donor-derived bone marrow cells were predominantly detected in the bone marrow, blood, and spleen of recipient mice six weeks after transplantation (S5B Fig). Furthermore, we also performed western blot analysis of RNF213 in chimeric mice, and found that RNF213 was notably detected in the spleen from *Rnf213⁻/⁻* mice that received bone marrow from WT mice, but not detected in the spleen from WT mice that received bone marrow from *Rnf213⁻/⁻* mice (S5C Fig). These data collectively indicate that the transplantation system is successful in the chimeric mice. Next, chimeric mice were intravenously infected with *A. baumannii,* and the bacterial loads in the spleen and brain were determined. Interestingly, WT mice that received bone marrow from either WT or *Rnf213⁻/⁻* mice exhibited a significantly lower bacterial load in the brain than did *Rnf213⁻/⁻* mice that received bone marrow from either WT or *Rnf213⁻/⁻* mice (Fig 3J), whereas the bone marrow cells isolated from WT and *Rnf213⁻/⁻* mice had a comparable capacity to regulate the bacterial burden in the brain in WT and *Rnf213⁻/⁻* mice (Fig 3J). However, the bacterial load in the spleen was comparable among these groups (Fig 3K), suggesting that nonimmune cells but not immune cells are the dominant players in the process by which RNF213 suppresses bacterial infection in the brain. Moreover, we intravenously injected EBD into all chimeric mice to evaluate its leakage into the brain. Notably, the leakage of EBD in *Rnf213⁻/⁻* mice that received bone marrow from either WT or *Rnf213⁻/⁻* mice was significantly greater than that in WT mice that received bone marrow from either WT or *Rnf213⁻/⁻* mice (S5D–S5F Fig). Consistent with these findings, the expression of Claudin-5 in the brains of WT mice that received bone marrow from either WT or *Rnf213⁻/⁻* mice was greater than that in the brains of *Rnf213⁻/⁻* mice that received bone marrow from either WT or *Rnf213⁻/⁻* mice (S5G and S5H Fig). Collectively, these results indicate that RNF213 in nonimmune cells plays important roles in maintaining BBB integrity and suppressing brain invasion during *A. baumannii* infection.

## RNF213 modulates blood–brain barrier integrity via IFN-I signaling

RNF213 deficiency impaired immune responses and increased the bacterial load of *A. baumannii* in the brain. Compared with WT mice, *Rnf213⁻/⁻* mice presented reduced gene expression of *Cxcl9, Cxcl10, Ifnb, Tnfa* and *Il1a* in the brain in response to *A. baumannii* infection (Fig 4A). In addition, the production of IFN-β in the brain was much lower in *Rnf213⁻/⁻* mice than in WT mice 20 hours after intravenous infection with *A. baumannii* (Fig 4B). To determine whether the regulation

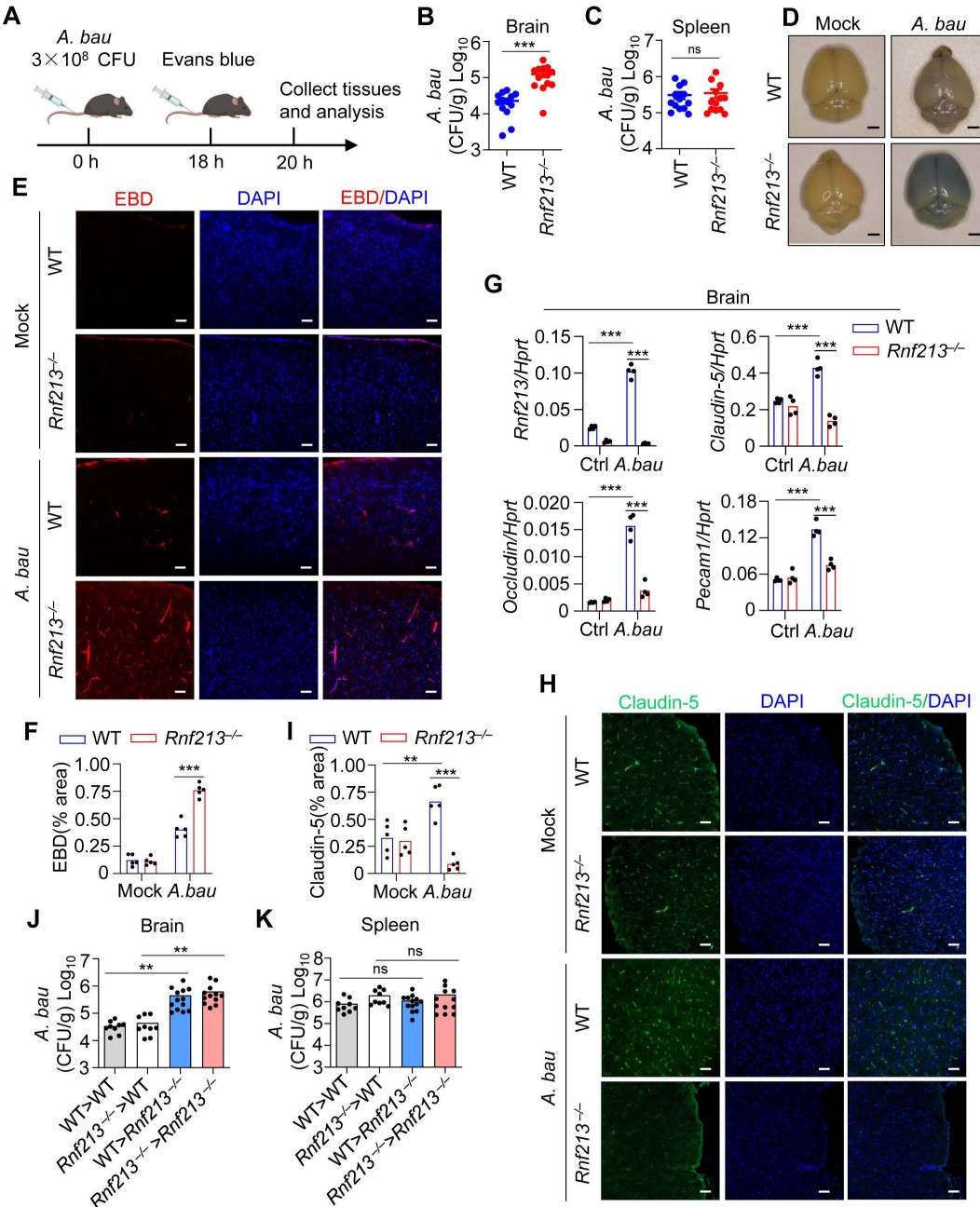

**Fig 3. RNF213 deficiency causes blood-brain barrier impairment. (A)** Strategy for intravenous *A. baumannii* infection and blood-brain barrier analysis in mice. Mice were intravenously infected with *A. baumannii* and colony-forming units (CFUs) were analyzed at 20 hours post infection. Evans blue dye (EBD) was intravenously injected in mice at 18 hours after infection and evaluated its leakage into the brain at 20 hours after infection. **(B-C)** WT (n = 13) and *Rnf213⁻/⁻* (n = 13) mice were intravenously infected 3.0 × 10⁸ CFU of *A. baumannii* per mouse. Bacterial burden in the brain (B) and spleen (C) at 20 hours after infection was measured. **(D-F)** Representative microscopy images (D) and confocal microscopy images **(E)**, and quantification analysis of EBD (F) showing vessel leakage in the brain of uninfected and *A. baumannii*-infected mice in **(B)**. (G) qRT-PCR analysis of *Rnf213, Claudin-5, Occludin,* and *Pecam1* in the brain of uninfected and *A. baumannii*-infected mice in **(B)**. (H-I) Representative confocal microscopy images (H) and quantification analysis (I) showing immunostaining for Claudin-5 in the brain of uninfected and *A. baumannii*-infected mice in **(B)**. **(J-K)** The gender- and age-matched WT > WT chimeras (n = 9), *Rnf213⁻/⁻* > WT chimeras (n = 9), WT > *Rnf213⁻/⁻* (n = 13) and *Rnf213⁻/⁻* > *Rnf213⁻/⁻* chimeras (n = 12) were intravenously infected 3.0 × 10⁸ CFU of *A. baumannii* per mouse. Bacterial burden in the brain (J) and spleen (K) at 20 hours after infection was measured. Data are from 3 independent experiments (B, C, J, K) or representative of 3 independent experiments with similar results (D, E, F, G, H, **I)**. Each symbol indicates an individual mouse for (B, C, G, J, **K)**. Each dot represents one field and total 5 random fields were quantified for (F) and **(I)**. Scale bar: 20 mm for **(D)**, 50 µm for **(E, H)**. Data represent Mean ± SEM for (B, C, F, G, I-K), 2-sided Student's *t*-test without multiple-comparisons correction, **$P < 0.01$; ***$P < 0.001$; ns, not significant. A was created with Biorender.com.

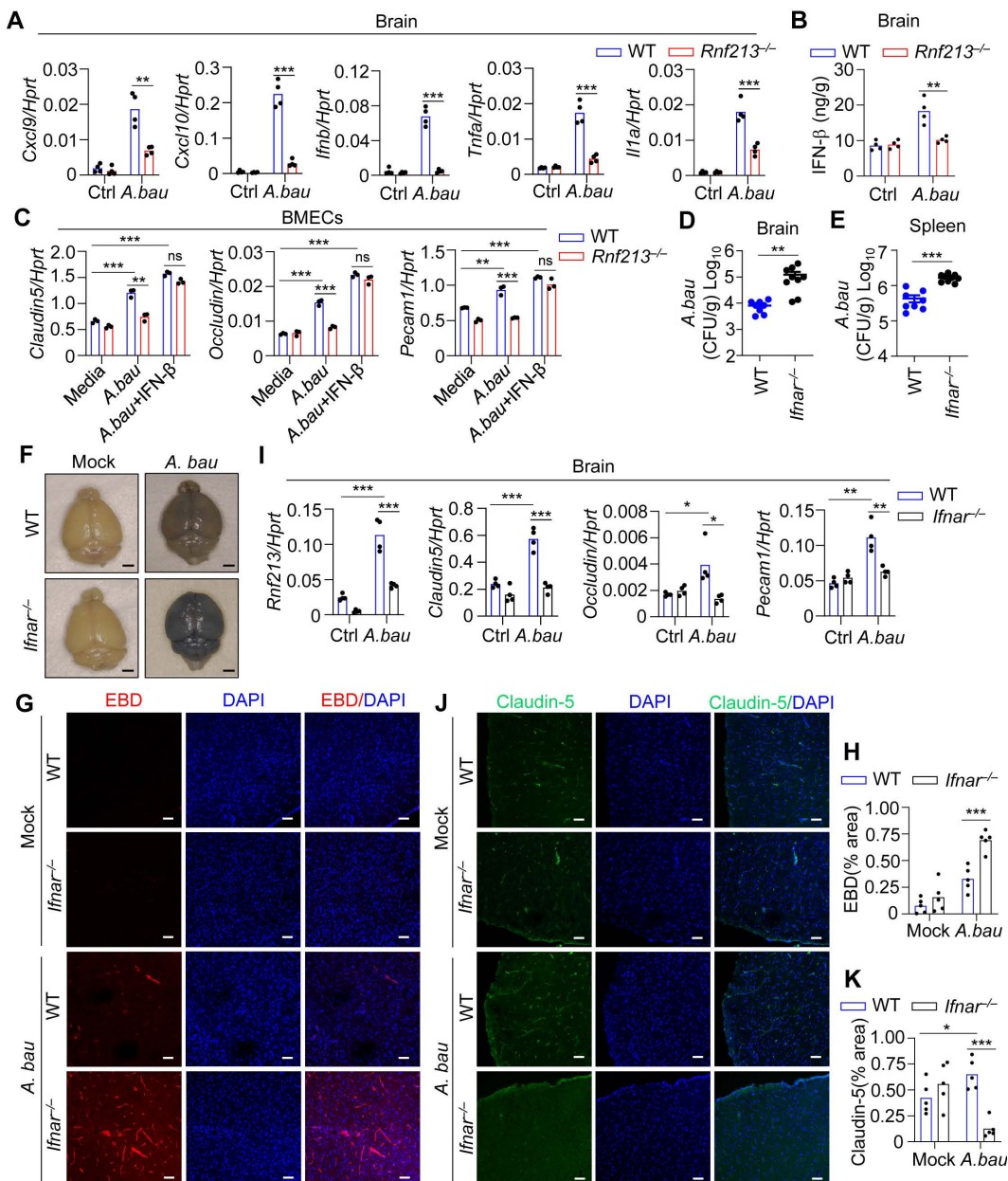

**Fig 4. IFN-I is required for maintaining blood-brain barrier integrity during *A. baumannii* infection.** (A-B) qRT-PCR analysis of *Rnf213, Cxcl9* and *Cxcl10* **(A)**, and ELISA analysis of IFN-β release (B) in the brain of uninfected and *A. baumannii*-infected mice (3.0 × 10⁸ CFU per mouse for 20 hours). (C) qRT-PCR analysis of *Claudin-5, Occludin,* and *Pecam1* in WT and *Rnf213⁻/⁻* BMECs without treatment or infected with *A. baumannii* (50 MOI, 6 **h)**, or pretreated with IFN-β (2 ng/mL) for 1 h and further stimulated with *A. baumannii* (50 MOI, 6 **h)** (n = 3 technical replicates; 3 independent experiments). **(D-E)** WT (n = 8) and *Ifnar⁻/⁻* (n = 9) mice were intravenously infected with *A. baumannii* (3.0 × 10⁸ CFU per mouse). Bacterial burden in the brain (D) and spleen (E) at 20 hours after infection was measured. **(F-H)** Representative microscopy images (F) and confocal microscopy images (G) and quantification analysis of EBD (H) showing vessel leakage in the brain of uninfected and *A. baumannii*-infected mice in **(D)**. (I) qRT-PCR analysis of *Rnf213, Claudin-5, Occludin,* and *Pecam1* in the brain of uninfected and *A. baumannii*-infected mice in **(D)**. **(J-K)** Representative confocal microscopy images (J) and quantification analysis (K) showing immunostaining for Claudin-5 in the brain of uninfected and *A. baumannii*-infected mice in **(D)**. Data are from 3 independent experiments (D-E) or representative of 3 independent experiments with similar results (A-C, F-K). Each symbol indicates an individual mouse for (A-B, D-E, **I)**. Each dot represents one field and total 5 random fields were quantified for H and **K.** Scale bar: 20 mm for **(F)**, 50 μm for **(G, J)**. Data represent Mean ± SEM for (A-E, I, H-K), 2-sided Student's *t*-test without multiple-comparisons correction, *P < 0.05; **P < 0.01; ***P < 0.001; ns not significant.

of the BBB by IFN-β is associated with the expression of TJ-related proteins, BMECs were isolated from WT mice and *Rnf213*^−/−^ mice and then stimulated with *A. baumannii* in the presence of IFN-β; the gene expression of endothelial TJ-related proteins was subsequently evaluated. Compared with WT BMECs, *Rnf213*^−/−^ BMECs exhibited significantly lower expression of *Claudin-5, Occludin, and Pecam1* in response to *A. baumannii* infection. Notably, the reduced expression of *Claudin-5, Occludin, and Pecam1* in *Rnf213*^−/−^ BMECs was reversed by treatment with IFN-β (Fig 4C).

To determine whether IFN-I signaling is required for BBB integrity in vivo, we determined the bacterial loads in the brain and spleen of WT and *Ifnar*^−/−^ mice that were intravenously infected with *A. baumannii.* Twenty hours after infection with *A. baumannii*, the bacterial loads in the brain and spleen were significantly greater in *Ifnar*^−/−^ mice than in WT mice (Fig 4D and 4E). We found that *A. baumannii* infection caused massive leakage of EBD throughout the brain in *Ifnar*^−/−^ mice, unlike in WT mice (Fig 4F). Microscopy further revealed that EBD fluorescence in the brain parenchyma was markedly greater in *Ifnar*^−/−^ mice than in WT mice after *A. baumannii* infection (Fig 4G and 4H). In addition, we measured the mRNA expression levels of endothelial TJ-related proteins in the brains of WT and *Ifnar*^−/−^ mice. In line with the results from *Rnf213*^−/−^ mice, the expression of *Claudin-5, Occludin, and Pecam1* following *A. baumannii* infection was significantly lower in *Ifnar*^−/−^ mice than in WT mice (Fig 4I–4K, S6A and S6B Fig). Collectively, these results indicate that RNF213 modulates blood–brain barrier integrity via IFN-I signaling.

## RNF213 targets TRAF3

RNF213 deficiency inhibited TBK1 phosphorylation and IFN-I production in response to LPS stimulation and *A. baumannii* infection but not in response to IFN-β treatment, suggesting that RNF213 positively regulates IFN-I signaling by acting on targets upstream of IFNAR (Figs 2D, 2E, S3C and S3D). Thus, coimmunoprecipitation (co-IP) experiments were performed to evaluate the interactions between exogenously expressed RNF213 and TRIF, TRAF3, TBK1, and IRF3, which are critical molecules for IFN-I production, and the results revealed that RNF213 specifically interacted with TRAF3 but not with TRIF, TBK1, or IRF3 in human embryonic kidney HEK293T cells (Figs 5A and S7A). The interaction between RNF213 and TRAF3 was also confirmed by reciprocal co-IP (Fig 5B). To confirm the interaction of endogenous RNF213 with TRAF3, we infected WT and *Rnf213*^−/−^ MSFs with *A. baumannii* and performed immunoprecipitation with an anti-TRAF3 antibody. Notably, our results revealed that RNF213 interacted with TRAF3 in control and *A. baumannii*-infected WT MSFs, and increased RNF213–TRAF3 interaction following *A. baumannii* infection might be partially due to elevated RNF213 expression levels (Fig 5C). Furthermore, the colocalization of RNF213 and TRAF3 in HEK293T cells was confirmed by confocal microscopy (Fig 5D). TRAF3 contains an N-terminal RING domain, a TRAF domain and a C-terminal MATH domain [2]. To determine which domain of TRAF3 is responsible for its interaction with RNF213, a series of TRAF3 truncation mutants were constructed. Co-IP experiments with anti-RNF213 antibody and these TRAF3 truncation mutants revealed that the N-terminal RING domain of TRAF3 is required for its interaction with RNF213 (Fig 5E). TRAF3 contains an N-terminal RING domain composed of seven conserved cysteines and one histidine, which are involved in the coordination of two atoms of zinc. Mutation of the last residue of the domain can compromise the stability of the adjacent cysteine, affecting the coordination of the zinc atom [27,34,35]. We replaced the last residue 91 glutamine (Q) within the domain of mouse TRAF3 with alanine (A) to generate the Q91A mutant of TRAF3 (S7B and S7C Fig). The Co-IP assay results revealed that the Q91A mutation blocked the interaction between RNF213 and TRAF3 (Fig 5F).

## RNF213 regulates the K27-linked polyubiquitination of TRAF3

Given that the E3 ligase RNF213 interacts with TRAF3 and positively regulates IFN-I activation, we investigated whether RNF213 regulates the function of TRAF3 by modulating its ubiquitination. Thus, after transfecting HEK293T cells with RNF213, TRAF3 and HA-tagged WT ubiquitin or K6-, K11-, K27-, K29-, K33-, K48-, or K63-specific ubiquitin, in which only one of the seven lysine residues was retained and the other lysine residues were mutated, we performed co-IP assays. RNF213 promoted TRAF3 polyubiquitination with HA-tagged WT ubiquitin and HA-tagged K27-specific ubiquitin,

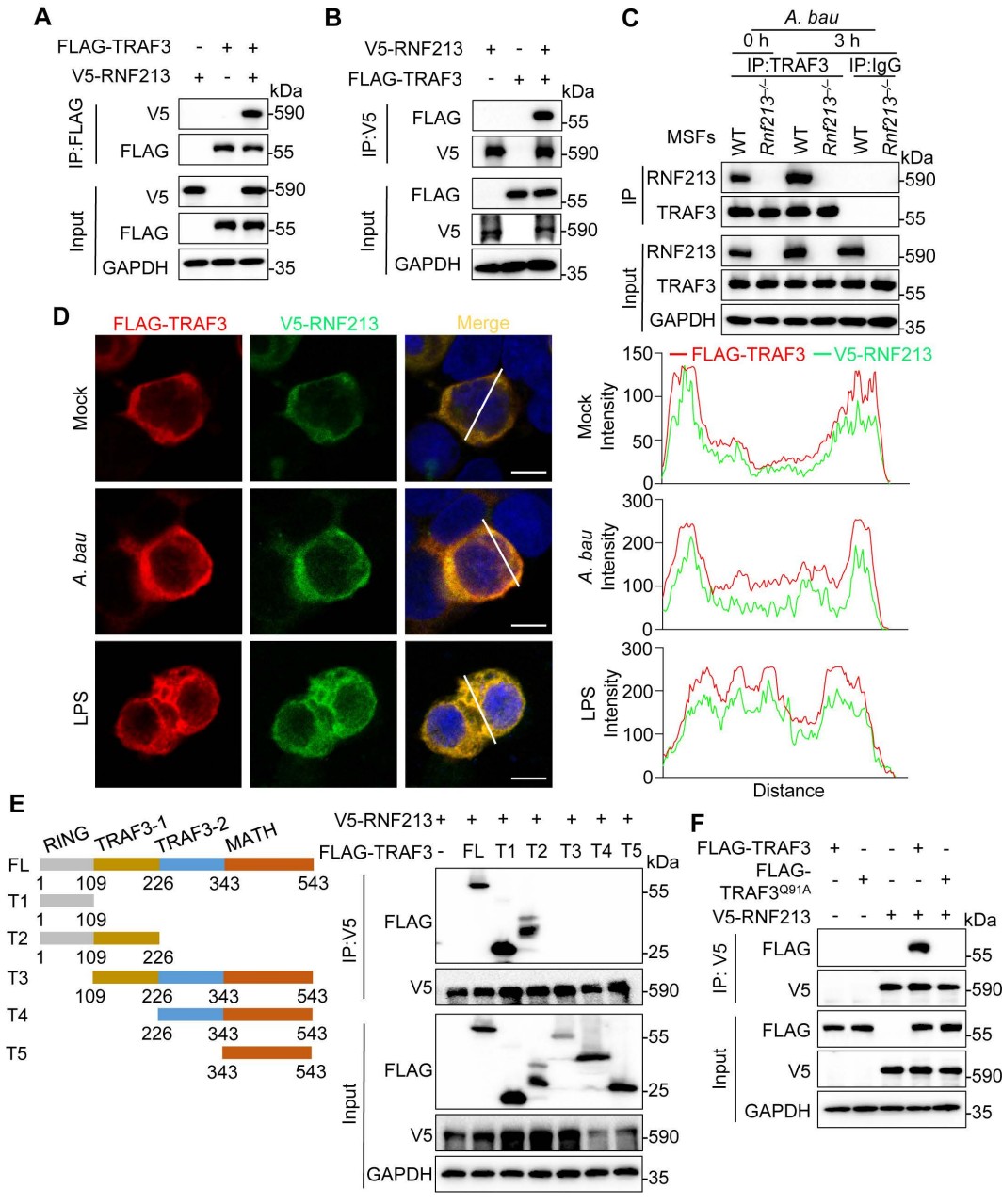

**Fig 5. RNF213 interacts with TRAF3. (A)** Immunoblot analysis of V5-RNF213 co-IP with FLAG-TRAF3 from lysates of HEK293T cells transfected with plasmids as indicated. **(B)** Immunoblot analysis of FLAG-TRAF3 co-IP with V5-RNF213 from lysates of HEK293T cells transfected with plasmids as indicated. **(C)** Co-IP analysis of endogenous TRAF3 interacting with RNF213 in WT and *Rnf213⁻/⁻* MSFs with or without *A. baumannii* (50 MOI) infection for 3 hours. **(D)** Representative confocal microscopic images of colocalization of V5-RNF213 and FLAG-TRAF3 in HEK293T cells with or without *A. baumannii* (50 MOI) infection or LPS (200 ng/mL) treatment for 3 hours. Scale bars: 10 μm. **(E)** Immunoblot analysis of V5-RNF213 co-IP with full length and truncated FLAG-TRAF3 from lysates of HEK293T cells transfected with plasmids as indicated. **(F)** Immunoblot analysis of V5-RNF213 co-IP with WT FLAG-TRAF3 and catalytic site (Q91A) mutant from lysates of HEK293T cells transfected with plasmids as indicated. Data are representative of 3 independent experiments with similar results **(A-F)**.

indicating that RNF213 increased mainly the K27-linked polyubiquitination of TRAF3 (Fig 6A). This result was further confirmed by experiments with the K27R mutant (in which only the Lys27 residue was mutated to Arg) of ubiquitin. As expected, the results of immunoprecipitation followed by immunoblot analysis indicated that RNF213 did not catalyze TRAF3 polyubiquitination in the presence of the K27R mutant of ubiquitin (Fig 6B).

To examine whether the E3 ligase activity of RNF213 contributes to the K27-linked polyubiquitination of TRAF3, we mutated mouse RNF213 W3974 site homologous with the amino acid W4024 within the human RNF213 RING domain, which is found to be mutated in MMD patients and essential for E3 ligase activity (S7D Fig) [26]. Notably, the W3974R mutation in RNF213 abolished TRAF3 polyubiquitination, indicating that W3974 plays important roles for E3 ubiquitin ligase activity (Fig 6C). To confirm the endogenous RNF213-mediated ubiquitination of TRAF3, we analyzed the endogenous polyubiquitination of TRAF3 in WT and *Rnf213⁻/⁻* MSFs in response to *A. baumannii* infection. Notably, the K27-linked ubiquitination of endogenous TRAF3 induced by *A. baumannii* infection was reduced in the absence of RNF213 (Fig 6D). We tested whether RNF213–TRAF3 heterodimerization might create a catalytically active RING domain by mutating the Q91 linchpin residue (Fig 5F) [36]. However, the Q91A mutation abolished RNF213–TRAF3 interaction, suggesting that heterodimerization might be required for TRAF3 ubiquitination, which needs further investigation. We subsequently attempted to identify the sites of RNF213-mediated ubiquitination in TRAF3. We replaced each of the eight lysine (K) residues in TRAF3 individually with arginine (R) to generate the K138R, K154R, K156R, K160R, K168R and K181R mutants of TRAF3 [7]. The Co-IP assay results revealed that only the K160R mutation partially blocked the RNF213-mediated ubiquitination of TRAF3 (Fig 6E). To investigate whether the RNF213-promoted K27-linked polyubiquitination of TRAF3 at K160 is essential for IFN-I production, we transfected WT and mutant TRAF3 (K156R, K160R) into *Traf3⁻/⁻* MSFs and evaluated ISGs and TJ-related proteins expression under RNF213 overexpression. The expression of *Ifnb, Cxcl9, Cxcl10, Claudin-5, Occludin*, and *Pecam1* but not *Tnfa* was significantly higher in *Traf3⁻/⁻* MSFs cotransfected with WT TRAF3 or K156R mutant of TRAF3 and RNF213 than in those cotransfected with K160R mutant of TRAF3 and RNF213 in response to *A. baumannii* infection (Figs 6F and S8). Consistently, the K160R mutation of TRAF3 impaired its ability to activate TBK1 and IRF3 compared to WT TRAF3 and K156R (Fig 6G). Collectively, our findings demonstrate that RNF213 functions as an E3 ubiquitin ligase and directly catalyzes the K27-linked polyubiquitination of TRAF3 at K160. In summary, RNF213 maintains BBB integrity and supports host defense by mediating TRAF3 ubiquitination and IFN-I signaling during *A. baumannii* infection (Fig 7).

## Discussion

The BBB, composed of specialized endothelial cells with tight junctions, actively protects the CNS through selective exclusion of blood-borne pathogens, macromolecules, and cellular components [37]. The destruction of the BBB by invasion by different types of bacteria or systemic inflammation can lead to or aggravate various CNS diseases [38,39]. Studies have shown that the degree of BBB damage in MMD patients is significantly greater than that in patients with other cerebrovascular diseases [23]. RNF213 is the first recognized as an important and common susceptibility gene for MMD [24], and the expression of *Rnf213* was significantly regulated by IFN-I signaling [29,30]. However, the function of RNF213 in host defense against CNS infection remains unclear. In this study, we defined RNF213 is sensitive to IFN-I signaling under basal and stimulated conditions, and as a positive regulator of IFN-I production during host immune responses. This positive feedback loop involving RNF213 is established through mediation of the K27-linked polyubiquitination of TRAF3 and activation of IFN-I signaling. In previous studies, RNF213 was identified as a regulator of cerebral endothelial integrity and barrier function, with no indication of its specific targets [33,40]. Our results show that RNF213 promotes blood–brain barrier integrity during *A. baumannii* infection by increasing the IFN-I-dependent expression of genes that regulate endothelial tight junctions.

Moyamoya disease is increasingly recognized as an immune-associated angiopathy, and infection-induced immune responses might function as a second hit to trigger the onset of moyamoya disease [28]. Recently, RNF213 was

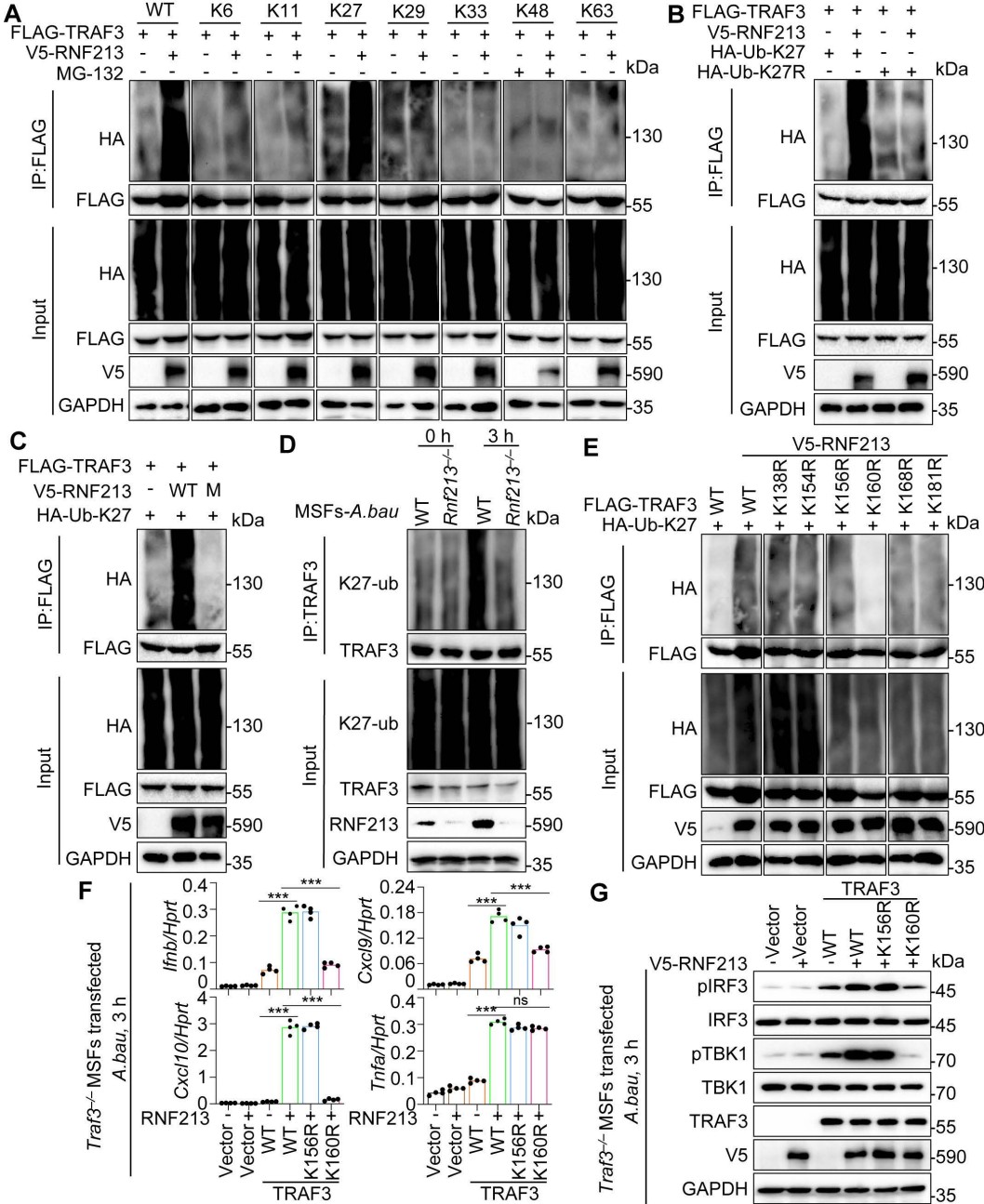

**Fig 6. RNF213 promotes the ubiquitination of TRAF3. (A)** Co-IP analysis of WT and K6-, K11-, K27-, K29-, K33-, K48-, or K63-linked polyubiquitination of FLAG-TRAF3 mediated by V5-RNF213 in HEK293T cells transfected with plasmids as indicated. MG132 (20 µM) was added to cells for 4 hours in K48-linked polyubiquitination analysis. **(B)** Co-IP analysis of WT K27- and K27 mutant (K27R)-linked polyubiquitination of FLAG-TRAF3 mediated by V5-RNF213 in HEK293T cells transfected with plasmids as indicated. **(C)** Co-IP analysis of the K27-linked polyubiquitination of FLAG-TRAF3 mediated by WT V5-RNF213 and catalytic site (W3974R) mutant in HEK293T cells transfected with plasmids as indicated. **(D)** Co-IP analysis the K27-linked polyubiquitination of endogenous TRAF3 in WT and *Rnf213⁻/⁻* MSFs in response to *A. baumannii* infection (50 MOI) for indicated times. **(E)** Co-IP analysis of the K27-linked polyubiquitination of WT FLAG-TRAF3 and its mutants K138R, K154R, K156R, K160R, and K168R by V5-RNF213 in HEK293T cells transfected with plasmids as indicated. (F) qRT-PCR analysis of *Ifnb, Cxcl9, Cxcl10,* and *Tnfa* in *Traf3⁻/⁻* MSFs transfected with RNF213 combined with WT or mutant TRAF3 plasmids (K156R and K160R), infected with *A. baumannii* (50 MOI) for indicated times. (n=4 technical replicates; 3 independent experiments). **(G)** Immunoblot analysis of phosphorylation of IRF3, TBK1, and IRF3, TBK1, TRAF3 and V5-RNF213 in *Traf3⁻/⁻* MSFs transfected with RNF213 combined with WT or mutant TRAF3 plasmids (K156R and K160R), infected with *A. baumannii* (50 MOI) for indicated times. Data are representative of 3 independent experiments with similar results **(A-G)**. Data represent Mean±SEM for **(F)**, 2-sided Student's *t*-test without multiple-comparisons correction, ***P<0.001; ns, not significant.

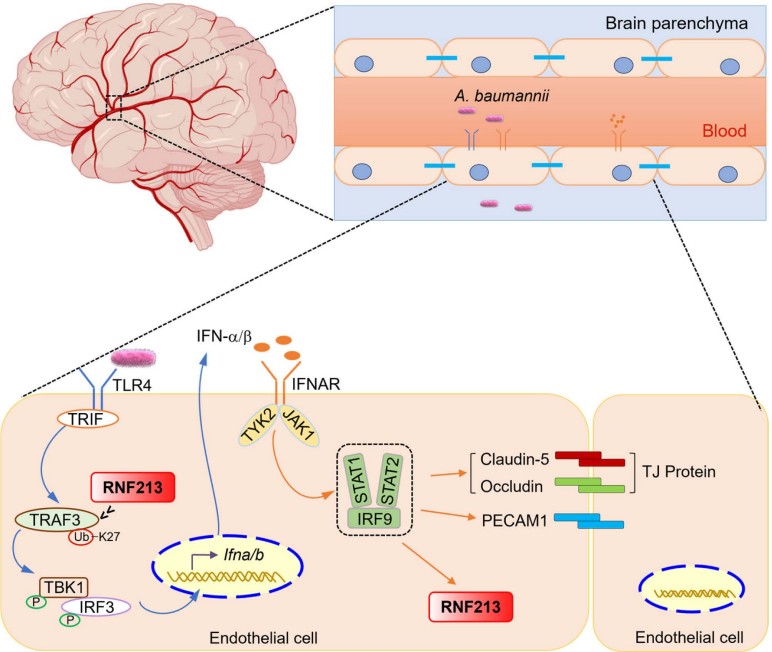

**Fig 7. Model of the mechanism by which RNF213 mediates TRAF3 ubiquitination and IFN-I activation in endothelial cells, which reinforce the BBB during *A. baumannii* infection.** Upon *A. baumannii* infection, TLR4 engagement results in the recruitment of the adaptor protein TRIF to activate TRAF3-dependent IFN-I signaling. RNF213 mediates the K27-linked polyubiquitination of TRAF3 and promotes type I IFN production. The expression of *Rnf213* is dependent on type I IFN signaling. RNF213 promotes the expression of *Claudin-5, Occludin,* and *Pecam1* and increases BBB integrity via IFN-I signaling. Created with Biorender.com.

investigated as a key antimicrobial protein that mediates noncanonical ubiquitylation of bacterial LPS for the initiation of antibacterial functions [16,41] and targets parasite-containing vacuoles and viral proteins for the execution of ubiquitylation-driven antipathogenic host defense responses [17,42,43]. To the best of our knowledge, TRAF3 is one of few identified host targets of RNF213 and mediates the expression of tight junction-regulating genes in brain microvascular endothelial cells, contributing to the maintenance of BBB integrity during microbial infection. Our findings elucidate the potential functions of the RNF213–mediated BBB integrity by targeting TRAF3 for the regulation of IFN-I signaling in moyamoya disease development.

Vascular endothelial cells serve as physiological barriers involved in many vascular physiological functions, such as permeability control and nutrient and oxygen transport, and in the elicitation of innate and adaptive immune responses [44,45]. Our bone marrow transplantation experiments revealed the crucial role of RNF213 within nonimmune cells in maintaining the integrity of the BBB and preventing bacterial infection in the brain. Moreover, we found that the reduced expression of *Claudin-5, Occludin, and Pecam1* in *Rnf213⁻/⁻* BMECs was rescued by treatment with IFN-β. We speculate that the initial IFN-I signaling within endothelial cells triggered by bacterial infection contributes to the activation of downstream cellular defense mechanisms and immune responses. Treatment with IFN-β protects the BBB in patients with relapsing–remitting multiple sclerosis (RRMS) [46]. The protective effect of IFN-β maintains the integrity of the endothelial cell barrier and counteracts the effects of barrier breaking factors such as the inflammatory factors TNF-α and IL-1β [47]. In addition, in vitro BBB models of brain capillary endothelial cells and rat astrocytes, IFN-β stabilized the integrity of the endothelial barrier [48]. However, we lack direct evidence that the compromised BBB permeability observed during *A. baumannii* infection in *Rnf213*-deficient mice results from inadequate IFN-β production by BMECs. The roles of RNF213 and IFN-I signaling in endothelial cells need to be further verified in mice with cell-specific conditional knockout. Despite

extensive validation of IFN-β's role in maintaining the BBB, the functional contributions of downstream ISGs in this context remain poorly understood and necessitate additional research.

TRAF3 promotes IFN-I production and inhibits inflammatory cytokine production typically through modification by different types of ubiquitination in response to infection or stimulation. The K33-linked polyubiquitination of TRAF3 at Lys168 was found to modulate the exocytosis of intracellular bacteria in infected BECs [49]. TRAF3 was found to be degraded in a manner dependent on its cIAP1/2- and TRAF6-mediated K48-linked polyubiquitination in response to MyD88-dependent TLR signaling. In contrast, TRIF-dependent signaling was found to induce the K63-linked polyubiquitination of TRAF3, which activates downstream IRF3 and the IFN-I response [50,51]. In this study, we demonstrate that the RING domain-mediated interaction between RNF213 and TRAF3 activates E3 ubiquitin ligase activity, suggesting that the RING heterodimerization between RNF213 and TRAF3 might create a catalytically active RING domain. RNF213 mediates the K27-linked polyubiquitination of TRAF3 and IFN-I signaling in BMECs, suggesting that TRAF3 has pleiotropic functions in IFN-I signaling pathway depending on the ubiquitination type and ubiquitination site within specific cell type.

In summary, our data reveal a previously unrecognized target in the mechanism by which RNF213 regulates the IFN-I response and establish a link between this MMD susceptibility gene and infectious diseases. Our study provides a deeper understanding of the function of RNF213 and reveals potential therapeutic targets against bacterial brain infection and moyamoya disease.

## Methods

### Ethics statement

All of the mice were maintained under specific-pathogen-free conditions, and all animal studies were approved by the Ethics Committee of Scientific Research of Shandong University (ECSBMSSDU2021-2-171), Jinan, Shandong Province, China.

### Mice

*Rnf213⁻/⁻* mice were generated by Cyagen Biosciences Inc. Exons 3–66 of the *Rnf213* gene were knocked out by CRISPR-Cas9 system. *Ifnar⁻/⁻* mice were previously described [10]. WT and knockout mice were SPF-clean and kept under specific pathogen-free conditions in the Animal Resource Center at Shandong University, Jinan, Shandong Province, China. All animal experiments were conducted in accordance with guidelines approved by the Ethics Committee of Scientific Research of Shandong University.

### Bacterial culture and infection of mice

The bacterial strain *A. baumannii* was growing as previously described [10]. Eight- to ten-week-old *Rnf213⁻/⁻* mice and WT controls were infected intravenously with *A. baumannii* ($3.0 \times 10^8$ CFUs per mouse) as indicated. Mice were weighted and monitored at day 0 and day 1. Mice were euthanized at days as indicated after infection and brain or spleen were harvested to determine the bacterial burden as described previously [52].

### Preparation of BMDMs, MSFs and BMECs for treatment, bacterial infection and siRNA Transfection

To generate BMDMs, bone marrow (BM) cells were cultured in L929 cell-conditioned DMEM/F-12 supplemented with 10% FBS, 1% non-essential amino acids and 1% Penicillin-streptomycin for 5 days. BMDMs were seeded in 12-well plates (1 million cells per well) and cultured overnight for use.

MSFs were prepared as previously described. Briefly, pinnae of adult wild type mice were minced and digested with 20 mg/mL collagenase type I (Sigma–Aldrich, C0130) for 3 h, followed by filtration through 70-μm strainers. Cells were cultured in 50% FBS in DMEM supplemented with HEPES, 1% penicillin and streptomycin, L-glutamine, sodium pyruvate,

non-essential amino acids, and β-mercaptoethanol for the first 3–4 days. Cells were subcultured in DMEM supplemented with 10% FBS and 1% penicillin and streptomycin for 2–3 days. MSFs were seeded in 12-well plates (0.2 million cells per well) and cultured overnight for use.

To generate BMECs, mice were euthanized, brains were collected in 10 mL of ice-cold PBS and meninges were removed by gently rolling the brains on sterile filter papers. Then the remaining brain tissues were cut into 1 mm³ blocks in DMEM/F-12. The brain was digested in 10 mL dispase solution [containing 5 mg/mL DNase I (Sangon Biotech, B002004-0005) and 10 mg/mL type II collagenase (Sangon Biotech, A004202)] for 60 min at 37 °C and the disrupted tissue was centrifuged at 800 × g at room temperature (RT) for 5 min. Supernatants were discarded, and the pellets was resuspended in 10 mL 20% bovine serum albumin and centrifuged at 1000 × g at RT for 20 min. After removal of nerve tissue in the upper layer, the blood vessels were digested with 3 mL of 0.125% trypsin at 37 °C for 30 min. After digestion at 37 °C, 5 mL of DMEM were added to the cell suspension, followed by 10 min centrifugation at 1,000g at RT. The microvessel fragments were washed once with DMEM/F-12 plus 10% FBS and centrifuged at 500 × g for 5 min. The cells were seeded into a fibronectin-coated culture flask and cultured in 10% FBS in DMEM/F-12 supplemented with ECGS (Sciencell, 1052), Ascorbate, L-glutamine, 1% penicillin-streptomycin and Heparin for 5–6 days. BMECs were seeded in 12-well plates (0.2 million cells per well) and cultured overnight for use.

The siRNAs target to *Rnf213* were ordered from RiboBio Company. siRNAs were electroporated into MSFs by using Neon TM Transfection System (Invitrogen, MPK10025) following manufacturer's instructions. The siRNA sequences are listed in S2 Table. WT, knockout, and siRNA-transfected MSFs were stimulated with ligands or infected with bacterial pathogens for indicated times as previously described [53]. The treated and control cells were lysed for RNA and protein analysis.

### In vivo assessment of BBB integrity

We evaluate BBB integrity in vivo using Evans blue dye (EBD) as a marker as previously described [54]. Mice were infected intravenously with *A. baumannii* as indicated and were intravenously injected with 2% EBD solution in saline (4 mL/kg). Then EBD was allowed to circulate for 2 hours. Whole-brain were removed and imaged using table operating microscope (SM-410D, YUYAN). To further evaluate EBD extracted from the brain, we imaged sections by fluorescence microscopy. Brains were fixed with 4% paraformaldehyde for 48 hours at room temperature and dehydrated in 15%-30% sucrose. After the specimens were frozen, 15 μm serial cryostat sections were made for fluorescence imaging. For immunofluorescent staining, the sections were incubated with appropriate antibodies (Claudin-5: Invitrogen, 34–1600, 1:300; Occludin: MCE, HY-P81231, 1:300). Images were acquired by a confocal microscope (LSM900, Zeiss) and processed by Image J software.

### Bone marrow transplantation

The mouse model of bone-marrow transplantation was established using WT (CD45.1⁺) and *Rnf213⁻/⁻* (CD45.2⁺) mice as the donors and the recipients, which were similar in age and within 8–12 weeks old [55,56]. Six to eight hours prior to transplantation, the recipient mice were irradiated with a dosage of 10 Gy. For transplantation, 10 million bone marrow cells were injected into the tail veins of the recipient mice. Since the second day of transplantation, recipients were fed antibiotics for two weeks. Six weeks after transplantation, bone marrow chimeric mice were used for infection experiments.

### Flow cytometry analysis

For flow cytometric analysis of CD45.1⁺ cells and CD45.2⁺ cells, cells prepared from spleens, bone marrow and peripheral blood were stained using a subset of antibodies (Biolegend, QA18A43, QA18A15, 1:300). Cell preparation and staining were carried out as described previously [57,58], and cells were analyzed on a BD LSRFortessa Cell Analyzer (BD Biosciences).

## Immunoblot analysis and antibodies

Samples were separated by 12% SDS-PAGE and 6% Tris-acetate-PAGE, followed by electrophoretic transfer to PVDF membranes, and membranes were blocked and then incubated with primary antibodies. The following primary antibodies were used: anti-IRF3 (Cell Signaling Technology [CST], 4302, 1:1000); anti-pIRF3 (CST, 4947, 1:1000); anti-TBK1 (CST, 3504, 1:1000); anti-pTBK1 (CST, 5483, 1:1000); anti-RNF213 (Millipore, ABC1391, 1:1000); anti-TRAF3 (CST, 4729, 1:1000); anti-FLAG (Sigma, F3165, 1:5000); anti-V5 (CST, 13202, 1:1000); anti-HA (ABclonal, AE008, 1:1000); anti-K27-ubiquitin (Abcam, ab181537, 1:1000); anti-IgG(CST, 2729, 1:1000) and anti-GAPDH (CST, 2118, 1:1000). HRP-labeled anti-rabbit, anti-mouse or anti-goat (CST) was used as the secondary antibody.

## Immunofluorescence staining

For V5-RNF213 and FLAG-TRAF3 immunostaining, HEK293T cells were seeded on poly-D-lysine coated coverslips in 12-well plates and transfected with indicated plasmids using lipofectamine 3000 reagents (Invitrogen, Thermo Fisher Scientific). 24 hours post-transfection, treated and untreated cells were washed with PBS and blocked in 1×ELISA buffer with 0.1% saponin for 1 hour. Cells were stained with anti-FLAG (Sigma, F3165, 1:500) and anti-V5 (CST, 13202, 1:500), overnight at 4 °C. Cells were washed, stained with a fluorescence conjugated secondary antibody for 60 min at 37 °C, and mounted using mounting medium (Vector Laboratories, H-1200). Cells were observed on the ZEISS-LSM900 confocal microscope, and data analysis were performed using Image J software.

## Plasmid transfection and co-IP experiments

The full-length of RNF213 and TRAF3 were amplified from a mouse cDNA library and subcloned into pCMV and pCDH vectors. Truncated DNA sequences were amplified from full-length of cDNA plasmids and subcloned into pCDH vector. Site-directed mutations were generated using QuikChange site-directed mutagenesis kits. All plasmids were confirmed by DNA sequencing [7]. Ubiquitin-expressing plasmids were generated as previously described [53]. The primer sequences for vector construction are listed in S2 Table. Lipofectamine 3000 reagents (Invitrogen, Thermo Fisher Scientific) were used for transient transfection of plasmids into HEK293T cells.

For IP, whole HEK293T cells collected 36 hours after transfection or MSFs with or without *A. baumannii* infection were lysed in IP buffer composed of 50 mM Tris-HCl (pH7.4), 2 mM EDTA, 150 mM NaCl, 1% NP-40, 10% Glycerol, 1 mM DTT, and protease/phosphatase inhibitor cocktails. After centrifugation, supernatants were collected and incubated with protein A/G Plus–Agrose (Santa Cruz Biotechnology, sc-2003) and 5 μg of the corresponding antibodies for 12 hours at 4°C, followed by washing 3 times with IP buffer. Anti-FLAG (Sigma, F3165), Anti-V5 (CST, 13202), Anti-RNF213 (Millipore, ABC1391), Anti-IgG (CST, 2729) and Anti-TRAF3 (CST, 4729) were used for endogenous co-IP analysis. Immunoprecipitated components were eluted by boiling in the SDS loading buffer for 10 min. For immunoblot analysis, immunoprecipitates and input lysates were separated by SDS–PAGE, followed by transferring onto PVDF membranes and detected by specific antibodies.

## Ubiquitination analysis

For polyubiquitination analysis of TRAF3 in HEK293T cells, HEK293T cells were transfected with plasmids expressing WT RNF213 or mutant RNF213 and HA-ubiquitin (WT), HA-ubiquitin (K6), HA-ubiquitin (K11), HA-ubiquitin (K27), HA-ubiquitin mutant (K27R), HA-ubiquitin (K29), HA-ubiquitin (K33), HA-ubiquitin (K63) or HA-ubiquitin (K48) with MG132, and Flag-TRAF3 (WT and mutants). For endogenous ubiquitination analysis of TRAF3 in MSFs, WT and *Rnf213*$^{-/-}$ MSFs were infected with *A. baumannii* for indicated times. Transfected HEK293T cells and MSFs were lysed in lysis buffer (50 mM Tris-HCl (pH 6.8), 1.5% SDS) and boiling for 15 min, the denatured samples were immunoprecipitated with the anti-FLAG and anti-TRAF3 antibodies, and the IP products were analyzed by immunoblot with anti-HA, anti-ubiquitin and specific antibodies.

### Real-time quantitative PCR

Total RNA was isolated from cells and tissues using TRIzol reagent (Invitrogen, Thermo Fisher Scientific). cDNA was reverse transcribed using M-MLV reverse transcriptase (Promega). Real-time quantitative PCR was performed on the Roche LightCycler 96 Real-Time Detection System. The primer sequences are listed in S2 Table. Hypoxanthine guanine phosphoribosyltransferase (HPRT) is a housekeeping gene and used as internal control as previously described [7].

### ELISA

The in vivo and in vitro samples were analyzed for cytokine release using ELISA MAX Standard Sets from BioLegend (Mouse TNF, 430901) and RnDSystems (Mouse IFN-β, DY8234-05) according to the manufacturer's instructions.

### Lentivirus production and infection

WT TRAF3, and TRAF3 mutants were cloned into the lentiviral expression vector pCDH-CMV-MCS-EF1-Puro. The viral particles were prepared by transfecting HEK293T cells with TRAF3-expressing or control plasmids in combination with packaging vectors. Twelve hours later, medium was replaced with fresh complete DMEM. Viral supernatant was harvested and passed through a 0.45 μm syringe filter at 36, 60, and 84 hours after transfection. To establish stably infected cells, *Traf3*$^{-/-}$ MSFs were infected with lentivirus as indicated in the presence of polybrene (8 μg/mL) 3 times. The infected cells were cultured in fresh media for at least 3 days prior to *A. baumannii* infection and analysis.

### Sequencing and data analysis

Total RNA was extracted from A. baumannii-infected and uninfected WT and *Ifnar*$^{-/-}$ BMDMs and subjected to commercial RNA-sequencing analysis (Novogene). Differential gene expression analysis was performed using the edge R package. First, genes with CPM (Counts Per Million) values greater than 10 in all samples were filtered, followed by TMM normalization to account for sequencing depth differences. Differential expression analysis was then conducted using the exact Test function, with a square-root-dispersion of 0.1. Genes with log2FoldChange ≥ 1 and p-value < 0.05 were classified as significantly upregulated, while those with log2FoldChange ≤ -1 and p-value < 0.05 were considered significantly downregulated. The volcano plot was generated based on the differential expression results using the ggplot2 package. Additionally, Gene Ontology (GO) enrichment analysis of the differentially expressed genes was performed using the R packages org.Mm.e.g.,db (version 3.18.0) and cluster Profiler (version 4.10.1). A heatmap was created to visualize the expression patterns of the differentially expressed genes using the pheatmap package (version 1.0.12).

### Statistical analyses

Statistical analysis and graph design were performed using GraphPad Prism software (Graphpad Software, version 8). Statistical significance between two groups was determined by unpaired two-tailed t test. Data are presented as the mean ± standard error of the mean (SEM). P-values of 0.05 or less were considered significant. Biological replicates as indicated in the figure legend.

### Supporting information

**S1 Fig. Differentially expressed genes in uninfected and *A. baumannii* infected WT and *Ifnar*–/– BMDMs.** (A) RNA sequencing analysis of gene expression in uninfected and *A. baumannii* infected WT and *Ifnar*$^{-/-}$ BMDMs for 3 hours. Differentially expressed genes in WT and *Ifnar*$^{-/-}$ BMDMs are shown. The volcano plots display the distribution of DEGs in WT and *Ifnar*$^{-/-}$ samples at 0 h (left) and 3 h (right) post infection. The dotted vertical lines indicate the fold-change threshold (±1 log$_2$FC), while the horizontal dotted line represents the significance threshold (P = 0.05). (B) Enrichment analysis of the signaling pathways that were highly expressed in WT BMDMs compared with *Ifnar*$^{-/-}$ BMDMs. (C) The top 40 genes

ranked differentially expressed genes in WT versus *Ifnar*$^{-/-}$ BMDMs under both uninfected and *A. baumannii*-infected conditions, with ranking determined by basal expression in uninfected WT cells.
(TIF)

**S2 Fig. RNF213 regulates type I IFN signaling in MSFs and HUVECs.** (A) qRT-PCR analysis of *Rnf213, Cxcl9, Cxcl10, Irf1, Tnfa,* and *Il1a* in WT MSFs transfected with RNF213 or Vector control, without treatment or infected with *A. baumannii* (50 MOI) for indicated times (n = 2 technical replicates; 3 independent experiments). (B) qRT-PCR analysis of *Rnf213, Cxcl9, Cxcl10, Irf1, Tnfa,* and *Il1a* in HUVECs transfected with RNF213 or Vector control, without treatment or infected with *A. baumannii* (50 MOI) for indicated times (n = 4 technical replicates; 3 independent experiments). (C) qRT-PCR analysis of *Rnf213, Cxcl9, Cxcl10, Irf1, Tnfa,* and *Il1a* in WT MSFs transfected with control siRNA or siRNAs specific to *Rnf213* without treatment or infected with *A. baumannii* (50 MOI) for indicated times (n = 2 technical replicates; 3 independent experiments). Data are representative of 3 independent experiments with similar results (A-C). Data represent Mean ± SEM for (A-C), 2-sided Student's *t*-test without multiple-comparisons correction, **$P < 0.01$; ***$P < 0.001$; ns, not significant.
(TIF)

**S3 Fig. RNF213 does not regulate NF-κB pathway.** (A-B) Targeting strategy used to generate *Rnf213*$^{-/-}$ mice (A) and genotyping of offspring generated from breeding of *Rnf213* heterozygous mice (B). (C-D) Quantification analysis of phosphorylation of IRF3 and TBK1, and total IRF3, TBK1, and RNF213 in WT and *Rnf213*$^{-/-}$ MSFs (C) and BMECs (D) without treatment or infected with *A. baumannii* (50 MOI), or stimulated with LPS (200 ng/mL) for indicated times (n = 3 independent experiments). (E-F) qRT-PCR analysis of *Tnfa, Il1a,* and *Il6* in WT and *Rnf213*$^{-/-}$ MSFs (E) and BMECs (F) without treatment or infected with *A. baumannii* (50 MOI) as indicated times (n = 2 technical replicates; 3 independent experiments). Data are from 3 independent experiments (C, D) or representative of 3 independent experiments with similar results (B, E-F). Data represent Mean ± SEM for (C-F), 2-sided Student's *t*-test without multiple-comparisons correction, **$P < 0.01$; ***$P < 0.001$; ns, not significant.
(TIF)

**S4 Fig. RNF213 regulates expression of Occludin following *A. baumannii* infection.** (A-B) Representative confocal microscopy images (A) and quantification analysis (B) showing immunostaining for Occludin in the brain of uninfected and *A. baumannii*-infected mice in Fig 3B. Data are representative of 3 independent experiments with similar results (A-B). Each dot represents one field and total 5 random fields were quantified for (B). Scale bar: 50 μm for (A). Data represent Mean ± SEM for (B), 2-sided Student's *t*-test without multiple-comparisons correction, ***$P < 0.001$.
(TIF)

**S5 Fig. Bone-marrow transplantation analysis between WT and *Rnf213*–/– mice.** (A) Strategy for the procedure of bone marrow transplantation and *A. baumannii* infection experiments. Six to eight hours prior to bone marrow transplantation, the recipient mice were irradiated with a dosage of 10 Gy for 10 minutes. For transplantation, 10 million bone marrow cells were injected into the tail veins of the recipient mice. Six weeks later, bone marrow chimeric mice were intravenously infected with *A. baumannii* and colony-forming units (CFUs) were analyzed at 20 hours after infection. Evans blue dye (EBD) was intravenously injected into mice at 18 hours after infection and leakage into the brain was evaluated 2 hours later. (B) Representative flow cytometry plots of CD45.1$^+$ and CD45.2$^+$ cells in bone marrow, peripheral blood and spleens from chimeric mice as indicated. (C) Immunoblot analysis of RNF213 in the spleen of chimeric mice. (D-F) Representative microscopy images (D) and confocal microscopy images (E), and quantification analysis of EBD (F) showing vessel leakage in the brain of uninfected and *A. baumannii*-infected mice assessed by EBD. (G-H) Representative confocal microscopy images (G) and quantification analysis (H) showing immunostaining for Claudin-5 in the brain of uninfected and *A. baumannii*-infected mice. Data are representative of 3 independent experiments with similar results (B-H). Each

dot represents one field and total 5 random fields were quantified for (F) and (H). Scale bar: 20 mm for (D), 50 μm for (E, G). Data represent Mean ± SEM for (F, H), 2-sided Student's *t*-test without multiple-comparisons correction, **$P < 0.01$; ***$P < 0.001$. A was created with Biorender.com.
(TIF)

**S6 Fig. IFN-I signaling regulates expression of Occludin following *A. baumannii* infection.** (A-B) Representative confocal microscopy images (A) and quantification analysis (B) showing immunostaining for Occludin in the brain of uninfected and *A. baumannii*-infected mice in Fig 4D. Data are representative of 3 independent experiments with similar results (A-B). Each dot represents one field and total 5 random fields were quantified for (B). Scale bar: 50 μm for (A). Data represent Mean ± SEM for (B), 2-sided Student's *t*-test without multiple-comparisons correction, ***$P < 0.001$.
(TIF)

**S7 Fig. RNF213 does not interact with TRIF, TBK1 or IRF3.** (A) Immunoblot analysis of V5-RNF213 co-IP with HA-TRIF, Myc-TBK1 and Myc-IRF3 from lysates of HEK293T cells transfected with plasmids as indicated. (B) Representation of the 'cross-brace' structure of the C3HC4 RING-finger domain, mediated through cysteine and histidine Zinc-binding. (C) Sequence alignment of mouse TRAF3 (mTRAF3, 51–91) and human TRAF3 (hTRAF3, 52–92) domains. (D) Sequence alignment of human RNF213 (hRNF213) RING and mouse RNF213 (mRNF213) RING domains. Data are representative of 3 independent experiments with similar results(A).
(TIF)

**S8 Fig. The K160R mutation of TRAF3 impaired the expression of *Claudin-5, Occludin*, and *Pecam1*.** qRT-PCR analysis of *Claudin-5, Occludin,* and *Pecam1* in *Traf3*$^{-/-}$ MSFs transfected with RNF213 combined with WT or mutant TRAF3 plasmids (K156R and K160R), infected with *A. baumannii* (50 MOI) for indicated times. (n = 4 technical replicates; 3 independent experiments). Data are representative of 3 independent experiments with similar results. Data represent Mean ± SEM, 2-sided Student's *t*-test without multiple-comparisons correction, **$P < 0.01$; ***$P < 0.001$.
(TIF)

**S1 Table. The total RNA-Seq dataset including differentially expressed genes in WT and *Ifnar*–/– BMDMs following infection with *A. baumannii*,** related to Figs 1 and S1.
(XLSX)

**S2 Table. Oligos used in this study.**
(XLSX)

## Acknowledgments

We thank Translational Medicine Core Facility of Shandong University for consultation and instrument availability that supported this work.

## Author contributions

**Conceptualization:** Xiaopeng Qi, Tao Xu.

**Data curation:** Yanfeng Li, Qingqing Xie, Luyu Yang, Xiaopeng Qi, Tao Xu.

**Formal analysis:** Yanfeng Li, Qingqing Xie, Luyu Yang, Xiaopeng Qi, Tao Xu.

**Funding acquisition:** Xiaopeng Qi, Tao Xu.

**Investigation:** Yanfeng Li, Qingqing Xie, Luyu Yang, Xiaopeng Qi.

**Methodology:** Yanfeng Li, Qingqing Xie, Luyu Yang, Xiaopeng Qi.

**Project administration:** Yanfeng Li, Qingqing Xie, Luyu Yang, Qian Jiang, Zhiping Liu, Chengjiang Gao, Xiaopeng Qi, Tao Xu.

**Resources:** Xiaopeng Qi, Tao Xu.

**Software:** Yanfeng Li, Qingqing Xie, Luyu Yang, Xiaopeng Qi.

**Supervision:** Xiaopeng Qi, Tao Xu.

**Validation:** Xiaopeng Qi, Tao Xu.

**Visualization:** Yanfeng Li, Xiaopeng Qi.

**Writing – original draft:** Yanfeng Li, Xiaopeng Qi, Tao Xu.

**Writing – review & editing:** Yanfeng Li, Qingqing Xie, Luyu Yang, Xiaopeng Qi, Tao Xu.

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
