## [Decision Letter · Decision Letter 0]

RNF213 regulates blood‒brain barrier integrity by targeting TRAF3 for type I interferon activation during A. baumannii infection

PLOS Pathogens

Dear Dr. Xiaopeng Qi,

Thank you for submitting your manuscript to PLOS Pathogens. Your manuscript has now been seen by three reviewers, who recognized the conceptual importance of your work and who commented positively on certain aspects of your manuscript while criticizing others. After careful consideration, we feel that your manuscript has merit but does not fully meet PLOS Pathogens's publication criteria as it currently stands. Therefore, we invite you to submit a revised version of the manuscript that addresses the points raised during the review process. Of particular importance are the missing transplantation controls highlighted by Reviewer 1 and 3 and experiments on the mechanism and consequences of ubiquitylation as suggested by all three reviewers.

Please submit your revised manuscript within 60 days, i.e. by April 15^th^ , 2025. If you will need more time than this to complete your revisions, please reply to this message or contact the journal office at plospathogens@plos.org. Please include the following items when submitting your revised manuscript:

We look forward to receiving your revised manuscript.

Kind regards,

Felix Randow

Guest Editor

PLOS Pathogens

Matthew Wolfgang

Section Editor

PLOS Pathogens

Editor-in-Chief

PLOS Pathogens

Michael Malim

Editor-in-Chief

PLOS Pathogens

orcid.org/0000-0002-7699-2064

**Journal Requirements:**

At this stage, the following Authors/Authors require contributions: Yanfeng Li, Luyu Yang, Qingqing Xie, Qian Jiang, Zhiping Liu, Chengjiang Gao, Xiaopeng Qi, and Tao Xu. Please ensure that the full contributions of each author are acknowledged in the "Add/Edit/Remove Authors" section of our submission form.

https://journals.plos.org/plospathogens/s/submission-guidelines#loc-parts-of-a-submission

4) We have noticed that you have uploaded Supporting Information files, but you have not included a complete list of legends. Please add a full list of legends for the supplementary tables after the references list. Please also include the Supplementary Figures in a separate file with the item type "Supplemental".

Potential Copyright Issues:

i) Please confirm (a) that you are the photographer of 3D, 4F, and S4B, or (b) provide written permission from the photographer to publish the photo(s) under our CC BY 4.0 license.

ii) Figures 3A.7, S2B, and S4A. Please confirm whether you drew the images / clip-art within the figure panels by hand. If you did not draw the images, please provide (a) a link to the source of the images or icons and their license / terms of use; or (b) written permission from the copyright holder to publish the images or icons under our CC BY 4.0 license. Alternatively, you may replace the images with open source alternatives. See these open source resources you may use to replace images / clip-art:

**Reviewers' Comments:**

Reviewer's Responses to Questions

**Part I - Summary**

Reviewer #1: Li et al. report that RNF213, previously known for its association with moyamoya disease, plays an important role in protecting the blood-brain barrier during Acinetobacter baumannii infection. Using a myriad of experimental approaches, they demonstrate that interferon signaling induces RNF213 expression, which then interacts with TRAF3 to mediate K27-linked polyubiquitination at K160. This molecular interaction promotes type I interferon signaling and regulates tight junction proteins including Claudin-5, Occludin, and Pecam1. They additionally demonstrate that the protective effects of RNF213 are specific to non-myeloid cells, where loss leads to compromised blood-brain barrier integrity and exacerbated A. baumannii infection. This study establishes a mechanistic link between RNF213 and blood-brain barrier function through TRAF3-mediated signaling, providing insights into both moyamoya disease pathogenesis and potential therapeutic strategies for CNS bacterial infections.

Reviewer #2: In this study the authors show how RNF213 deficiency impairs type I IFN production and decreases expression of ISGs in response to Acinetobacter infection. They reveal that RNF213 is required for efficient TBK1 and IRF3 phosphorylation upon infection with Acinetobacter or stimulation with LPS and further reveal that RNF213 interacts with TRAF3, but not associated signaling molecules TRIF, TBK1 or IRF3.

Analysis of TRAF3 ubiquitylation suggests that RNF213 mediates ubiquitylation at K160.

RNF213 is required to restrict the burden of Acinetobacter in the brain of infected mice (but not the spleen) and this is associated with a loss of blood brain barrier in the RNF213 KO.

The data is well-presented and by and large well controlled. The findings are of interest given the recent discovery that RNF213 is an immune sensor that controls numerous and diverse pathogens. The findings are significant.

Reviewer #3: Moyamoya disease is a cerebrovascular disease that can require surgery and be fatal. RNF213 is a major susceptibility gene, yet disease penetrance is extremely low. This suggests an additional, potentially environmental, factor contributes to the disease. The recently established role of RNF213 in microbial resistance implies pathogen infection might be connected, but a basis for this is unknown. The findings in this study suggest RNF213 plays a key role in BBB integrity during infection, providing a potential link to the aetiology of Moyamoya. This is an important finding. Moyamoya patient Mutations are found throughout the protein but cluster in the RING domain. However, the RING domain lacks E3 activity as it can be deleted with no defect in autoubiquitination. It also lacks a key linchpin residue required for RING E3 activity. Hence, the function of the RING domain, and its role in Moyamoya, remain unclear. The finding that the RING domain protein TRAF3 interacts with RNF213 and becomes ubiquitinated might explain the function of the RNF213 RING domain, but further experimentation is required (see below)

Overall, the paper is poorly written and lacks important citations throughout the results section. Without these its not clear what findings are novel. Indeed, some are claimed as such when they are not e.g. it is already known that RNF213 is an ISG. Figure legends are insufficient to appreciate what experiment is being performed and the methods section is not comprehensive enough to allow reproduction of the experiments.

Some aspects of the paper are very strong and convincing, but some require extensive revision or should be omitted entirely from the paper (RNA seq data).

**Part II – Major Issues: Key Experiments Required for Acceptance**

Reviewer #1: 1. The authors claim RNF213 expression is "dependent" on IFNAR signaling (lines 37-38), but the data show reduced but not absent expression in IFNAR knockout cells. The terminology should be modified to "partially dependent" or "significantly regulated by" to more accurately reflect the data.

2. The mechanism for RNF213 in directly ubiquitinating targets in IFN-I signaling is unclear. Including ubiquitination experiments to identify direct substrates of RNF213 or further explore pathways altered by RNF213 deficiency in the RNA-seq, may be helpful in discerning this mechanism.

3. The bone marrow transplantation experiments (Fig. 3J-K) require additional controls. The authors should include data demonstrating successful engraftment, chimerism quantification, and include flow cytometry data and analysis of the immune cell populations from chimeric mice.

4. The mechanism by which RNF213-mediated K27-linked ubiquitination of TRAF3 leads to increased type I interferon production is not fully elucidated. Additional experiments addressing the functional consequences of this modification on the activity or stability of TRAF3 are necessary.

5. Using macrophages derived from Rnf213 deficient mice to generalize the findings to human health is not convincing. Validation of this conclusion in a human cell line would better demonstrate the translatability of this data.

Reviewer #2: The main gap regarding the model and study conclusions is that figure 5 lays out that RNF213 and TRAF3 interact with each other, and that TRAF3 is ubiquitylated at K160, yet whether mutation of K160, so that it can’t be ubiquitylated, impacts downstream phenotypes including TBK1/IRF3 phosphorylation upon Acinetobacter infection, IFN/ISG production and expression of tight junction proteins is not analysed. This is important, as TRAF3 KO’s are not analysed either.

Related to this, whilst an RNF213 mutant is tested in the ubiquitylation of TRAF3 assay, unless a TRAF3 catalytic mutant is also analysed, does it not remain possible that TRAF3 mediates its own ubiquitylation, perhaps upon interaction or modification by RNF213?

Then, given sentences such as this “Our findings elucidate the potential functions of the RNF213–IFN-I–BBB axis in moyamoya disease development.” I would have expected the analysis of RNF213 mutations associated in moyamoya disease to have been analysed for TRAF3 ubiquitylation, IFN signalling and tight junction formation.

Reviewer #3: Initially, RNA seq data showing genes upregulated in WT vs IFNAR KO cells are related to innate immune responses, which is to be expected so there is nothing new presented here. There are also multiple issues with the presentation of the RNA-seq data.

• Media (Fig. S1A) and 0 hour treatment (Fig. S1B) are referred to and the numbers of upregulated proteins are different for both analyses. Are these different datasets? This is not explained properly.

• Figure 1A heatmap is incomprehensible. What is the reference for the relative fold changes (x/y)? Is y a reference protein? is it an earlier time point?? This lack of precision is unacceptable. Presumably the fold change is on a log2 scale but this is not specified. The colour ramp (red or blue) is also inappropriate to reveal changes the authors might be referring to.

Figure 1B RT-PCR data shows time dependent increase in RNF213 expression for a range of infection models and is strongly dependent on IFNAR. Legend should state what Hprt is. I am assuming this is a housekeeping protein used as a reference. Western blot data nicely validates the RT-PCR data. However, this is not surprising considering RNF213 is a known ISG. This should be stated along with an appropriate reference.

siRNA experiments then show that RNF213 is required for interferon signalling by measuring a decrease in the markers Cxcl9, Cxcl10 and Irf1. It should be stated what these markers represent, and a reference is once again required. The inflammatory cytokines Tnfa and II1a were unaffected by RNF213 KD so it is asserted that RNF213 mediates its effects upstream of IFNAR signalling.

S2 legends are once again inadequate and should state that the x axis corresponds to the time following infection, if that is what it is. Statistical tests used and P value assigned to *** should also be stated.

Due to the link between RNF213 with Moyamoya disease, bacterial load in the brain is measured in WT and RNF213 KO mice and is found elevated in the brain but not in the spleen, suggestive of organ-specificity. Bone marrow cross transplantation experiments are next carried out. This experiment needs to be more clearly explained. The below is also very difficult to follow:

“Interestingly, WT mice that received bone marrow from either WT or Rnf213–/– mice exhibited a significantly lower bacterial load in the brain than did Rnf213−/− mice that received bone marrow from either WT or Rnf213–/– mice (Fig 3J), whereas the bone marrow cells isolated from WT and Rnf213–/– mice had a comparable capacity to regulate the bacterial burden in the brain in WT and Rnf213–/– mice

(Fig 3J)”.

This experiment lacks important controls demonstrating bone marrow has been completely destroyed as the findings could be explained by residual marrow in WT samples following radiation treatment. This caveat should be clearly stated.

Strikingly, a compromised blood brain barrier is seen in RNF213 KO samples, as inferred from leakage of Evans blue dye into the organ. Quantitative confocal microscopy shows that baumanni infection enhances EBD leakage and this is exacerbated in RNF213 KO samples.

To test the hypothesis that brain leakage is due to RNF213 having a role in regulating TJ-related proteins, their levels (Claudin-5, Pecam1, Occludin) are measured by RT-PCR in response to infection. Analysis shows that baumannii infection upregulates TJ markers and this is suppressed in RNF213 KO cells. It is not clear if upregulation of TJ proteins upon infection of WT cells is statistically significant. P values should be assigned to the plots.

The microscopy does not show a clear increase in Claudin-5 in WT vs WT infected samples. The significance of this should also be calculated. Couldn’t the microscopy experiment be repeated with Occludin, which is stimulated more robustly upon infection?

Compromised expression of TJ markers upon infection is also observed in brains from RNF213 KO animals.

Similar brain phenotypes are observed with IFNAR KO mice. However, the statement that the findings “indicate that RNF213 modulates blood-brain barrier integrity via IFN-I signalling” is overstated and should be toned down.

Section RNF213 targets TRAF3

Where are the data showing RNF213 deficiency inhibits TBK1 phosphorylation? This is an essential piece of data. Following from this, co-IPs in HEK cells with a panel proteins involved in IFN production are carried out and TRAF3 is found to interact. Truncations clearly show that the N-terminal 109 residues of TRAF3 containing the RING is sufficient for interaction. The co-IP is also observed with endogenous proteins. Although for this experiment a control demonstrating that the TRAF3 antibody doesn’t cross react with RNF213 is missing. RNF213 levels are higher in baumanii infected cells so this might also explain why the apparent co-IP is higher. Microscopy data do support colocalization in cells.

It is next investigated whether RNF213 ubiquitinates TRAF3. Apparent ubiquitination of TRAF3 is observed and by using a panel of single lysine HA-Ub mutants, the data suggest K27-linkages predominate. A reverse experiment with a K27R mutant substantially reduces ubiquitination, further supporting modification with K27 linkages. To determine if RNF213 ligase activity is responsible a “catalytic site mutant (M)” is tested. There is no reference to the origin of this mutant, but the legend states it is a W3974R mutation. This is located in the RING domain. The authors must state how this mutation was chosen. Moreover, the RING domain does not contribute to RNF213 activity which depends on its RZ domain where it uses a non-RING transthiolating mechanism (Ahel et al. Biorxiv). The RING domain also lacks a hydrophilic linchpin residue required for RING E3 activity. Thus, the function of the RING domain remains unclear but is clearly relevant to Moyamoya disease because disease mutations cluster within it. Whether TRAF3 in isolation is an active E3 is also unclear. RING domains often require homo or heterodimerisation (e.g. BRCA1-BARD1) to become active. An exciting possibility is that to become active, the RNF213 RING domain heterodimerises with the TRAF3 RING domain, which does have a functional Gln linchpin (human Gln92). The authors should test if Gln92, or the equivalent tryptophan mutation in TRAF3 is required, in addition to RNF213, for TRAF3 ubiquitination.

**Part III – Minor Issues: Editorial and Data Presentation Modifications**

Reviewer #1: 1. The Western blot in Figure 2D requires quantification across multiple experiments with appropriate statistical analysis.

2. In Figure 4, the authors should include size markers on all immunofluorescence images and provide more detailed information about image acquisition and processing in the methods section.

3. Lines 353-355: The sentence beginning with "In this study, we define RNF213..." is unclear.

4. The discussion section (starting line 352) would be strengthened by acknowledging the study limitations more directly and discussing other known regulators of the blood-brain barrier and how they may correlate to the provided mechanism.

5. The methods for primary BMEC isolation and culture, the criteria for quantifying BBB disruption, and details for how the RNA-seq was analyzed could be expounded upon.

6. Several figure legends lack sufficient detail about experimental conditions and numbers of replicates.

7. Supplementary Table 1 should include adjusted p-values for the RNA-seq analysis.

8. Throughout the manuscript, the authors inconsistently use "A. baumannii" and "A.bau." This should be standardized.

9. For Figures 2B and 2C, error bars should be included for all time points.

10. The RT-qPCR data should include information about the housekeeping genes used and how they were validated for these experimental conditions.

11. Statistical analyses should be more clearly described in the methods section, including specific tests used for each type of comparison.

Reviewer #2: Figure 5A – the IP of RNF213 by TRAF3 shows some RNF213 coming down without any TRAF3, implying a degree of stickiness. Ideally a different tagged protein would be used for the “bait” in the IP assay in both this figure and 5B.

Figure 5C – not hugely convinced on the interaction becoming enhanced with infection. If RNF213 and TRAF3 interact in steady state, what is causing the induction of the signal upon infection/LPS?

Figure 5D - Microscopy appears to show the protein everywhere in the cell, perhaps as it is heavily overexpressed. From this condition, then, it doesn’t seem possible to discern whether the proteins are really colocalising and as one can’t tell which cells are infected, whether this is something induced by infection.

The text should make clear which catalytic mutant of RNF213 is used in figure 6 for analysis of TRAF3 ubiquitylation. The figure legend states it is W3974R but it should be made clear in the text if this is the RING domain or the non-canonical RZ finger in the E3 shell. The fact it has two domains that mediate ligase activity should also be mentioned in the introduction and discussion. This is important regarding the claim that TRAF3 is the first host substrate of RNF213.

Reviewer #3: (No Response)

PLOS authors have the option to publish the peer review history of their article (what does this mean? ). If published, this will include your full peer review and any attached files.

**Do you want your identity to be public for this peer review?** For information about this choice, including consent withdrawal, please see our Privacy Policy .

Reviewer #1: No

Reviewer #2: No

Reviewer #3: No

**Figure resubmission:**

**Reproducibility:**



---

## [Decision Letter · Decision Letter 1]

PPATHOGENS-D-24-02349R1

RNF213 regulates blood‒brain barrier integrity by targeting TRAF3 for type I interferon activation during A. baumannii infection

PLOS Pathogens

Dear Dr. Qi,

Thank you for submitting your manuscript to PLOS Pathogens. After careful consideration, we feel that it has merit but does not fully meet PLOS Pathogens's publication criteria as it currently stands. Therefore, we invite you to submit a revised version of the manuscript that addresses the points raised during the review process.

Please submit your revised manuscript within 30 days Aug 03 2025 11:59PM. If you will need more time than this to complete your revisions, please reply to this message or contact the journal office at plospathogens@plos.org. Please include the following items when submitting your revised manuscript:

We look forward to receiving your revised manuscript.

Kind regards,

Felix Randow

Guest Editor

PLOS Pathogens

Matthew Wolfgang

Section Editor

PLOS Pathogens

Sumita Bhaduri-McIntosh

Editor-in-Chief

PLOS Pathogens

orcid.org/0000-0003-2946-9497

Michael Malim

Editor-in-Chief

PLOS Pathogens

orcid.org/0000-0002-7699-2064

**Additional Editor Comments:**

Dear Dr Xiaopeng Qi,

Your revised manuscript has now been seen by the three original reviewers. All acknowledge the substantial efforts you have made to address their initial comments.

However, two reviewers have remaining concerns that must be addressed before the manuscript can be accepted for publication. Many of these points can likely be resolved through a more detailed discussion in the manuscript. That said, I would like to emphasize that Reviewer 2’s concerns regarding the potential lack of specificity of a key reagent in your study—the TRAF3 antibody—will require convincing experimental evidence to be fully resolved.

We look forward to receiving your revised manuscript.

Best regards,

Felix Randow

**Reviewers' Comments:**

Reviewer's Responses to Questions

**Part I - Summary**

Reviewer #1: The authors have performed considerable new experimentation and addressed all of my prior comments. I have no further suggestions. The manuscript is exciting and will be of interest to a wide readership.

Reviewer #2: The authors have added a substantial amount of data to the manuscript addressing many of the major concerns.

Reviewer #3: The authors have made satisfactory efforts to improve the paper. Some elements remain overstated and lack objective discussion.

**Part II – Major Issues: Key Experiments Required for Acceptance**

Reviewer #1: Not applicable

Reviewer #2: I have only one remaining query related to figure 5C - reviewer comments noted that to demonstrate a well controlled IP of RNF213 with TRAF3 requires further controls, e.g. to demonstrate that TRAF3 antibody isn't non-specifically bringing down RNF213 in the experiment - the provided blot using a TRAF3 antibody on input samples does not seem to address this and it is perhaps important given that a band of the molecular weight of RNF213 is present in the RNF213 KO IP lanes.

Reviewer #3: The following should be considered before acceptance:

Figure 1A – Please label the color ramp with “normalized Z-score” for clarity.

Results – IFN-I regulation of Rnf213:

The sentence:

"Taken together, these results indicate that IFN-I mediates the increase in Rnf213 expression induced by pathogen infection and stimuli�which is consistent with previous study that RNF213 as one of ISGs [29, 30]."

could be more clearly phrased. Suggested revision:

“Taken together, these results indicate that type I interferon (IFN-I) mediates the pathogen-induced upregulation of Rnf213, consistent with previous studies identifying Rnf213 as an interferon-stimulated gene (ISG) [29, 30].”

Results – TBK1 Phosphorylation:

Thank you for highlighting the TBK1 phosphorylation data. To improve clarity for the reader, please reference these data again in the “RNF213 Targets TRAF3” section. i.e.:

“RNF213 deficiency inhibited TBK1 phosphorylation and IFN-I production in response to LPS stimulation and A. baumannii infection, but not in response to IFN-β treatment, suggesting that RNF213 positively regulates IFN-I signaling by acting on targets upstream of IFNAR (Fig 2D and 2E, S3C and S3D).”

RNF213 Interaction with TRAF3:

It should be mentioned that the observed increase in RNF213–TRAF3 interaction following A. baumannii infection may be, at least in part, due to elevated RNF213 expression levels.

RING Dimerization and E3 Activity:

The statement:

“Furthermore, the homo- or hetero-dimer RING domain-interaction confers E3 ubiquitin ligase activity [2].”

is unclear.

Hu et al. (Ref 2) do not demonstrate that homodimerization of BRCA1 or BARD1 results in functional E3 activity. Therefore, the possibility that RNF213 requires heterodimerization for E3 activity remains valid.

The authors should state that they tested whether RNF213–TRAF3 dimerization is required for E3 ligase activity by mutating the Q91 linchpin residue. This discussion would be more appropriate in the “RNF213 regulates the K27-linked polyubiquitination of TRAF3” section.

Given that the Q91 mutation abolished interaction, it is unfortunate this prevented formal testing. Nonetheless, it should be clearly stated that heterodimerization may still be required for TRAF3 ubiquitination, and the discussion should reflect this possibility more objectively.

W3974R Mutation and E3 Activity:

The manuscript still asserts that W4024 (W3974 in human RNF213) is essential for E3 ligase activity. However, this conclusion may be overstated.

• Guey et al. (Ref 1) speculate that the W4024R mutation impairs activity based on analogy to other RING E3s but do not directly test RNF213 activity.

• Garcia-Barcena et al. is a review article and does not discuss RNF213 specifically.

• Takeda et al. (Ref 3) assess the W4024R variant using a GST-RING fusion, which may artificially enhance activity due to GST dimerization. Although they observe modest activation of Ubc13/Uev1a, the physiological relevance is unclear. Moreover, NF-κB activation was seen even with E3-dead mutants, including W4024R.

Thus, it should be stated that W3974R is predicted to be essential for E3 activity, which is what Ref 26 (used to support the claim) asserts.

**Part III – Minor Issues: Editorial and Data Presentation Modifications**

Reviewer #1: Not applicable

Reviewer #2: I would like to see further discussion regarding TRAF3 RING mutant Q91A and lack of interaction to RNF213 - does it imply TRAF3 ligase activity is required for stable interaction? I also find this sentence vague "In this study,

we show that the interaction between RNF213 and TRAF3 through RING domain may ignite the E3 ubiquitin ligase activity [51], and RNF213 ..." are the authors suggesting that the RING heterodimerisation between RNF213 and TRAF3 might create a catalytically active RING domain and if so, this should be stated more clearly.

Reviewer #3: (No Response)

PLOS authors have the option to publish the peer review history of their article (what does this mean? ). If published, this will include your full peer review and any attached files.

**Do you want your identity to be public for this peer review?** For information about this choice, including consent withdrawal, please see our Privacy Policy .

Reviewer #1: No

Reviewer #2: No

Reviewer #3: No

**Figure resubmission:**
---

## [Editor Report · Decision Letter 2]

Dear Dr Qi,

We are pleased to inform you that your manuscript 'RNF213 regulates blood‒brain barrier integrity by targeting TRAF3 for type I interferon activation during A. baumannii infection' has been provisionally accepted for publication in PLOS Pathogens.

Best regards,

Felix Randow

Guest Editor

PLOS Pathogens

Matthew Wolfgang

Section Editor

PLOS Pathogens

Sumita Bhaduri-McIntosh

Editor-in-Chief

PLOS Pathogens

orcid.org/0000-0003-2946-9497

Michael Malim

Editor-in-Chief

PLOS Pathogens

orcid.org/0000-0002-7699-2064

Dear Dr Tao Xu,

Thank you for the latest revision of your manuscript. I am pleased to inform you that all reviewers' comments have been satisfactorily addressed, and I now recommend your study for publication.

Thank you again for submitting your work to PLoS Pathogens.

With kind regards,

Felix Randow
---

## [Editor Report · Acceptance letter]

Dear Dr Qi,

We are delighted to inform you that your manuscript, "RNF213 regulates blood‒brain barrier integrity by targeting TRAF3 for type I interferon activation during A. baumannii infection," has been formally accepted for publication in PLOS Pathogens.

Best regards,

Sumita Bhaduri-McIntosh

Editor-in-Chief

PLOS Pathogens

orcid.org/0000-0003-2946-9497

Michael Malim

Editor-in-Chief

PLOS Pathogens

orcid.org/0000-0002-7699-2064